# FIDIA: Function-Informed Sequence Design via Inference-Aligned Policy Optimization

Minghan Li [1 2]   Fengji Li [1 2]   Yilin Tao [1 2]   Yue Deng [1 2]

## Abstract

Computational protein design typically employs a sequential workflow of structure generation followed by sequence (re)design. While structure generators can be explicitly conditioned on functional objectives, inverse folding models are constrained by their function-agnostic nature and sequence-structure degeneracy. More critically, the associated training objectives do not account for the *Best-of-N* (BoN) inference protocol, resulting in a fundamental training-inference misalignment. Here, we propose **FIDIA**, a reinforcement learning framework that enables **F**unction-**I**nformed sequence **D**esign via **I**nference-**A**ligned policy optimization. Specifically, FIDIA integrates functional constraints into composite rewards and explicitly optimize the induced policy under BoN toward high-fitness sequence regions. We achieve this via a grounded gradient estimator that directly maximizes the expected maximum reward. FIDIA consistently outperforms both standard and RL-optimized baselines in success rate and precision on a general motif scaffolding benchmark. Further experiments on realworld cases including vaccine and affinity-enhancing enzyme design validate FIDIA's efficacy in complex therapeutic and biocatalytic contexts.

## 1. Introduction

A fundamental principle of structural biology posits that sequence instructs the fold, and the fold orchestrates the function (Anfinsen, 1973; Albanese et al., 2025). Inverting this hierarchy, computational protein design predominantly employs a two-stage pipeline: (i) generating a function-conditioned backbone, modeled as $P(\mathbf{backbone} \mid$ **function**) (Ahern et al., 2025; Geffner et al., 2025b; Ingraham et al., 2023); and (ii) sampling an amino acid sequence onto this fixed backbone using an inverse folding model, approximated as $P(\mathbf{sequence} \mid \mathbf{backbone})$ (Hsu et al., 2022; Dauparas et al., 2025; Gao et al., 2023). Recent end-to-end approaches aim to directly approximating the joint distribution $P(\mathbf{sequence}, \mathbf{backbone} \mid \mathbf{function})$ (Qu et al., 2024; Lu et al., 2025; Geffner et al., 2025a). Nevertheless, employing a distinct sequence redesign step remains essential to resolve local geometric conflicts and maximize sequence-structure self-consistency (Butcher et al., 2025).

However, the efficacy of this standard workflow is fundamentally constrained by the generic inverse folding models (Qiu et al., 2024; Norn et al., 2021). Typically optimized to maximize Sequence Recovery Rate (Dauparas et al., 2022; Hsu et al., 2022; Dauparas et al., 2025), these models implicitly impose an assumption of conditional independence, approximating the ideal distribution $P(\mathbf{sequence} \mid \mathbf{backbone}, \mathbf{function})$ as merely $P(\mathbf{sequence} \mid \mathbf{backbone})$. This approximation ignores the inherent sequence-structure degeneracy (Ferrada, 2014), where numerous sequences may adopt the target backbone without meeting its functional requirements. Crucially, target functionality is highly sensitive to local physicochemical nuances, such as the geometric integrity of epitopes in vaccine design (Zhao et al., 2025; Huang et al., 2025) and the precise configuration of side-chain chemistries in enzyme engineering (Huang et al., 2025). Sampling from such an unconditioned distribution fails to account for these fine-grained specificities, often yielding sequences that are structurally viable yet functionally compromised, thereby undermining the success of the entire protein design pipeline.

The challenge of functional design is further compounded by a fundamental disconnect between the Maximum Likelihood Estimation(MLE) training objective and the *Best-of-N* (BoN) inference protocol. Specifically, BoN reflects the practical pipeline of protein engineering, where a large batch of candidates is generated for *in silico* screening, with only the top performers progressing to subsequent costly *in vivo* experiments. While MLE optimizes the overall probability mass of the learned distribution, functional utility hinges on the high-fitness tail behavior, resulting in a critical objec-

[1]School of Artificial Intelligence, Beihang University, Beijing, China [2]Zhongguancun Academy, Beijing, China. Correspondence to: Yue Deng <ydeng@buaa.edu.cn>.

*Proceedings of the 43rd International Conference on Machine Learning*, Seoul, South Korea. PMLR 306, 2026. Copyright 2026 by the author(s).

tive mismatch. Recent efforts (Xu et al., 2025; Wang et al., 2025) have attempted to inject functional guidance into inverse folding models leveraging various reinforcement learning frameworks. However, these approaches primarily focus on universal physical properties, failing to account for the specific functional requirements of individual proteins. Furthermore, the training-inference misalignment remains unexplored within the protein design pipeline.

In this work, we propose **FIDIA**(**F**unction-**I**nformed Sequence **D**esign via **I**nference-**A**ligned Policy Optimization), a framework that empowers inverse folding models to satisfy complex biological constraints under the *Best-of-N* inference protocol. Specifically, we reformulate sequence generation into a reward-driven process, using ESMFold (Lin et al., 2023) and VinaDock (Eberhardt et al., 2021) as oracles to provide task-specific feebback. We focus on directly optimizing the peak functional utility of sampled batches to ensure strict alignment between training and inference-time strategies. To this end, we employ a theoretically derived estimator that circumvents the non-differentiability and sample inefficiency of the max operator. Empirically, we evaluate our framework on sequence design for vaccine motif scaffolding and affinity-enhancing enzyme scaffolding. The results demonstrate that FIDIA significantly outperforms both standard and BoN-optimized RL baselines, consistently achieving superior motif consistency and binding potency.

Our contributions are summarized as follows:

- We align the objective of established inverse folding models using reinforcement learning, shifting the generative focus from maximizing sequence recovery to identifying sequences with superior task-specific functional properties.

- We theoretically derive a gradient estimator to directly optimize the *Best-of-N* objective, effectively aligning the training process with the inference-time protocol.

- FIDIA elevates the success rate from 26.67% to 37.50% and achieves the lowest motif RMSD on general motif scaffolding benchmarks. Furthermore, it demonstrates state-of-the-art performance in vaccine and affinity-enhancing enzyme design, notably stabilizing the Vinadock score from -6.53 to -7.21.

## 2. Related work

**Inverse Folding Models**   Inverse folding involves generating amino acid sequences that fold into a predefined three-dimensional backbone structure. Deep learning-based methods typically formulate this task as a graph-to-sequence problem, aiming to maximize the native sequence recovery rate (Dauparas et al., 2022; Gao et al., 2023; Qiu et al., 2024). Recent works have attempted to impose physical priors onto

inverse folding models to enhance biological plausibility. Specifically, Xu et al. (2025), Cao et al. (2025), Xue et al. (2025), and Zhu et al. (2026) utilize Direct Preference Optimization (DPO) (Rafailov et al., 2024), whereas Wang et al. (2025) leverages Group Relative Policy Optimization (GRPO) (Shao et al., 2024) to improve generation quality. However, these methods primarily focus on optimizing universal physical constraints such as structural consistency and general physicochemical properties, often neglecting the functional requirements essential for specific protein design tasks. Other reinforcement learning paradigms applied to biomolecule design remain orthogonal to our scope, as they target unconditioned de novo generation (Mistani & Mysore, 2024), focus on mutation optimization (Subramanian et al., 2024; Sun et al., 2025), or require training auxiliary reward models and value networks (Angermueller et al., 2019).

***Best-of-N*: From Inference to Training**   Inference-time augmentation has emerged as an effective strategy for enhancing model performance   (Liu et al., 2025; Yao et al., 2023; Wang et al., 2023), with Best-of-N (BoN) sampling being widely adopted for its simplicity (Cobbe et al., 2021; Rakhsha et al., 2025).   Recent research increasingly focuses on integrating BoN strategies directly into the training phase. Representative approaches fine-tune the policy on self-generated, top-ranked samples, effectively distilling the inference-time search capability into the model parameters (Touvron et al., 2023; Deng et al., 2020; Bai et al., 2022). More advanced methods seek to minimize the distributional divergence between training and inference-time BoN   (Chow et al., 2024; Verdun et al., 2025). However, these approaches often incur high sample complexity, necessitating extensive sampling during training.

## 3. Methods

### 3.1. Problem Formulation

**Inverse Folding as an MDP**   With the primary goal of steering the generation of sequences toward superior functional fitness, we adopt a reinforcement learning framework and formulate the protein design process as a Markov Decision Process (MDP), aligning with previous approaches in the field (Xu et al., 2025; Wang et al., 2025). In this paper, we specifically focus on ProteinMPNN (Dauparas et al., 2022), which autoregressively samples amino acid sequences following a random decoding order.

Formally, we define a protein sequence of length $L$ as $\mathbf{s} = (r_1, \ldots, r_L)$, where each token $r_i \in \mathcal{V}$ represents the amino acid residue type at the $i$-th position, and $\mathcal{V}$ denotes the vocabulary of the 20 standard amino acids. Accounting for the random decoding order inherent in ProteinMPNN, we let $\mathcal{U} = (u_1, \ldots, u_L)$ denote the decoding order, where the

residue at position $u_t$ is determined at time step $t$.

We define the key components of the MDP tuple $(\mathcal{S}, \mathcal{A}, \pi, \mathcal{T}, R)$ as follows:

- **State** $S_t$: The state $S_t = (\mathbf{x}, c, r_{u_{1:t-1}})$ comprises the global context (backbone geometry $\mathbf{x}$ and function-related condition $c$) and the partial sequence determined at the preceding decoding positions $u_{1:t-1}$.

- **Action** $a_t$: An action $a_t \in \mathcal{V}$ represents the specific amino acid type assigned to the residue at the current decoded position $u_t$.

- **Policy** $\pi_\theta$: Parameterized by $\theta$, the policy $\pi_\theta$ defines a categorical distribution over the vocabulary $\mathcal{V}$. It specifies the probability of selecting residue $a_t$ for position $u_t$ given the current state, denoted as $\pi_\theta(a_t \mid S_t)$.

- **Transition** $\mathcal{T}$: The deterministic transition updates the sequence state by setting $r_{u_t} \leftarrow a_t$. The subsequent state $S_{t+1} = (\mathbf{x}, c, r_{u_{1:t}})$ includes this newly determined residue.

- **Reward** $R$: We define a terminal reward function $R(\mathbf{s}, \mathbf{x}, c)$ to evaluate the quality of the fully generated sequence $\mathbf{s}$. This non-differentiable function quantifies the functional fitness of the sequence while accounting for its structural compatibility with the backbone $\mathbf{x}$ (details in AppendixC.5).

**Objective aligned with Inference Protocal** Given the resource-intensive nature of wet-lab validation, researchers typically employ a *Best-of-N* (BoN) inference strategy: generating a batch of candidates and selecting the top-performers via *in silico* validation for experimental characterization. Formally, we define a *candidate set* $\mathbf{s}_{1:N} = \{\mathbf{s}_i\}_{i=1}^N$ as a set of $N$ sequences independently sampled from the policy, i.e. $\mathbf{s}_i \overset{\text{i.i.d.}}{\sim} \pi_\theta(\cdot \mid \mathbf{x}, c)$, and the probability of each sequence is factorized as $\pi_\theta(\mathbf{s} \mid \mathbf{x}, c) = \prod_{t=1}^L \pi_\theta(r_{u_t} \mid S_t)$. To facilitate mathematical derivation without loss of generality, we simplify the BoN selection to the single best sequence. Consequently, the effective performance of a computational design pipeline is dictated specifically by the peak performance of the candidate that yields the highest reward within the inference-time candidate set:

$$\mathbf{s}^* = \arg\max_{\mathbf{s} \in \mathbf{s}_{1:N}} R(\mathbf{s}, \mathbf{x}, c). \tag{1}$$

However, the standard reinforcement learning objective is to maximize the expected return of the generated sequences, which implicitly optimizes the average performance of the candidate set:

$$J_{\text{std}}(\theta) = \mathbb{E}_{\mathbf{s}} [R(\mathbf{s}, \mathbf{x}, c)]$$
$$\approx \mathbb{E}_{\mathbf{s}_{1:N}} \left[ \frac{1}{N} \sum_{i=1}^N R(\mathbf{s}_i, \mathbf{x}, c) \right]. \tag{2}$$

This standard formulation reveals a fundamental misalignment between the training objective and the practical requirements of protein engineering. By maximizing the expectation, the policy tends to concentrate probability mass around the mode of the reward landscape to minimize variance, often resulting in a conservative, risk-averse distribution. In contrast, discovering novel functional proteins requires a policy that promotes heavy tails towards the high-fitness regions, a necessity corroborated by methods like Evolve-Pro (Jiang et al., 2024) through their focus on high-fitenss exploration.

To bridge this gap, we shift the optimization objective to explicitly maximize the expected reward of the best candidate within the candidate set $\mathbf{s}_{1:N}$. By characterizing the best candidate as a sample drawn from an induced distribution $\mathbf{s}^* \sim \pi_{bon}(\mathbf{s} \mid \mathbf{x}, c; \theta)$, the objective transforms from the expected return to the expected maximum return:

$$J_{\text{BoN}}(\theta) = \mathbb{E}_{\mathbf{s}^*} [R(\mathbf{s}^*, \mathbf{x}, c)]$$
$$\approx \mathbb{E}_{\mathbf{s}_{1:N}} \left[ \max_{i=1,\dots,N} R(\mathbf{s}_i, \mathbf{x}, c) \right]. \tag{3}$$

Crucially, directly optimizing Eq. (3) aligns the training signal with the inference goal. This incentivizes the policy to concentrate probability mass on the upper tail of the reward distribution, thereby facilitating the discovery of protein designs with peak functional fitness. This formulation introduces two primary optimization hurdles: the non-differentiability of the max operator, which invalidates standard policy gradient techniques, and extreme sample inefficiency due to gradient reliance on a single top-ranked sequence. We address these challenges in the following sections by deriving a tractable gradient and developing a data-efficient optimization framework.

### 3.2. Derivation of the BoN Policy Gradient

In this section, we derive the analytical policy gradient for the BoN objective by characterizing the score function of the induced distribution $\nabla_\theta \ln \pi_{bon}(\mathbf{s}^*)$ in terms of the base policy $\pi_\theta$.

Treating the reward $R(\mathbf{s})$ as a random variable $r$, we first define its Cumulative Distribution Function (CDF) under the base policy as $F_\theta(r) = \mathbb{P}_{\mathbf{s} \sim \pi_\theta}[R(\mathbf{s}) \leq r]$, which quantifies the probability that a trajectory sampled from $\pi_\theta$ yields a reward no greater than $r$. Accordingly, the probability density of an arbitrary trajectory $\mathbf{s}$ under the induced distribution

$\pi_{bon}$ is given by:

$$\pi_{bon}(\mathbf{s}) = N \cdot \pi_\theta(\mathbf{s}) \cdot [F_\theta(R(\mathbf{s}))]^{N-1}. \qquad (4)$$

Intuitively, this density represents the probability that a specific trajectory $\mathbf{s}$ is generated within a set of $N$ i.i.d. samples and identified as the top-ranked candidate $\mathbf{s}^*$.

Applying the log-derivative trick to Eq. (3) and incorporating the factorization of $\pi_{bon}$ from Eq. (4) yields:

$$\begin{aligned} &\nabla_\theta J_{\text{BoN}} \\ &= \mathbb{E}_{\mathbf{s}^*} \left[ R(\mathbf{s}^*) \nabla_\theta \ln \pi_{bon}(\mathbf{s}^*) \right] \\ &= \mathbb{E}_{\mathbf{s}^*} \left[ R(\mathbf{s}^*) \Big( \nabla_\theta \ln \pi_\theta(\mathbf{s}^*) + (N-1) \nabla_\theta \ln F_\theta(R(\mathbf{s}^*)) \Big) \right]. \end{aligned} \qquad (5)$$

We proceed by characterizing the log-CDF gradient $\nabla_\theta \ln F_\theta(r)$ as a conditional expectation over the base policy $\pi_\theta$, as established in the following proposition:

**Proposition 3.1.** *(Proof in Appendix A.1) The gradient of the log-CDF is equivalent to the expectation of the truncated base policy's score function at the reward upper-bound $r$:*

$$\nabla_\theta \ln F_\theta(r) = \mathbb{E}_{\mathbf{s} \sim \pi_\theta(\cdot \mid R(\mathbf{s}) \leq r)} \left[ \nabla_\theta \ln \pi_\theta(\mathbf{s}) \right]. \qquad (6)$$

By substituting the identity established in Prop. 3.1 back into the expansion in Eq. (5), we resolve the CDF term and arrive at the final analytical expression for the BoN policy gradient:

$$\begin{aligned} \nabla_\theta J_{BoN} = \mathbb{E}_{\mathbf{s}^*} \Bigg[ R(\mathbf{s}^*) \Big( &\underbrace{\nabla_\theta \ln \pi_\theta(\mathbf{s}^*)}_{\text{Score of } \mathbf{s}^*} \\ &+ (N-1) \underbrace{\mathbb{E}_{\mathbf{s} \sim \pi_\theta(\cdot \mid R(\mathbf{s}) \leq R(\mathbf{s}^*))} [\nabla_\theta \ln \pi_\theta(\mathbf{s})]}_{\text{Expected Score of the Associated Sub-level Set}} \Big) \Bigg]. \end{aligned} \qquad (7)$$

This analytical expression characterizes a unique collaborative optimization mechanism: the first term shifts policy mass toward the peak performer $\mathbf{s}^*$, while the second term—counter-intuitively—reinforces the sub-level ensemble using the same advantage signal $R(\mathbf{s}^*)$. This reflects a probabilistic necessity: to maximize the likelihood of a candidate emerging as the "best" within a set, the optimization must collectively elevate both the specific winner and its underlying supporting distribution.

### 3.3. Monte Carlo Gradient Estimation

To estimate the gradient derived in Eq. (7), we seek to establish unbiased Monte Carlo estimators for the nested expectations: the inner expectation over $\mathbf{s} \sim \pi_\theta(\cdot \mid R(\mathbf{s}) \leq R(\mathbf{s}^*))$ and the outer expectation over $\mathbf{s}^* \sim \pi_{bon}$.

To tractably estimate the inner expectation, we utilize the order statistics of the candidate set $\mathbf{s}_{1:N}$. Let $\{\mathbf{s}_{(i)}\}_{i=1}^N$ denote the samples reindexed such that $R(\mathbf{s}_{(1)}) \leq \cdots \leq R(\mathbf{s}_{(N)})$, where $\mathbf{s}_{(N)}$ serves as a proxy for the optimal sample $\mathbf{s}^*$.

Specifically, we consider the estimation of the term:

$$\eta_i = \mathbb{E}_{\mathbf{s} \sim \pi_\theta(\cdot \mid R(\mathbf{s}) \leq R(\mathbf{s}_{(i)}))} \left[ \nabla_\theta \ln \pi_\theta(\mathbf{s}) \right]. \qquad (8)$$

For a given $\mathbf{s}_{(i)}$, the subset of preceding samples $\{\mathbf{s}_{(j)}\}_{j=1}^{i-1}$ provides an empirical approximation of the conditional distribution $\pi_\theta(\cdot \mid R(\mathbf{s}) \leq R(\mathbf{s}_{(i)}))$. Consequently, an empirical estimator $\hat{\eta}_i$ can be constructed via the sample mean:

$$\hat{\eta}_i = \frac{1}{i-1} \sum_{j=1}^{i-1} \nabla_\theta \ln \pi_\theta(\mathbf{s}_{(j)}). \qquad (9)$$

The required inner expectation for our optimization objective can thus be effectively approximated by $\hat{\eta}_N$.

Building on this inner expectation estimator, we further estimate the outer expectation. We first simplify the notation as $\mu = \mathbb{E}_{\mathbf{s}^*}[f(\mathbf{s}^*)]$, where $f$ denotes the complete analytical term within the expectation. A naive Monte Carlo estimator would approximate $\mu$ via the empirical mean of $f(\mathbf{s}^*)$, where each sample $\mathbf{s}^* \sim \pi_{bon}$ is obtained by drawing $N$ samples from the base policy and selecting the one with the maximum reward.

While theoretically unbiased, this approach is statistically inefficient. By treating sub-optimal samples as mere sub-level sets to estimate the conditional score function for the current winner, it ignores their potential to lead in different batches. Crucially, it overlooks the rich combinatorial information available across different batches, resulting in high estimation variance and poor data efficiency. To mitigate these issues, we propose a data-efficient estimator that leverages a larger batch $\mathcal{B}_K$ i.e. $s_{1:K}, K > N$ to significantly reduce variance, achieving a provably tighter bound.

**Proposition 3.2.** *(Proof provided in Appendix A.2) An unbiased estimator leveraging the full batch $\mathcal{B}$ is given by:*

$$\hat{\mu} = \sum_{i=1}^K \frac{\binom{i-1}{N-1}}{\binom{K}{N}} f(\mathbf{s}_{(i)}) = \sum_{i=1}^K p_i f(\mathbf{s}_{(i)}), \qquad (10)$$

*where $p_i = \binom{i-1}{N-1} / \binom{K}{N}$ quantifies the probability that the $i$-th sorted trajectory $\mathbf{s}_{(i)}$ emerges as the Best-of-N candidate within a randomly sampled subset of size $N$ (with $\binom{n}{k} = 0$ for $n < k$).*

*For comparison, consider the standard estimator $\hat{\mu}_{std}$ constrained to the same sample budget $K$. This estimator is statistically equivalent to averaging over a random partition of $\mathcal{B}_K$ into $M = \lfloor K/N \rfloor$ disjoint subsets $\{\mathcal{B}_j\}_{j=1}^M$:*

$$\hat{\mu}_{std} = \frac{1}{M} \sum_{j=1}^M f\left( \arg\max_{\mathbf{s} \in \mathcal{B}_j} R(\mathbf{s}) \right). \qquad (11)$$

*Crucially, the proposed estimator $\hat{\mu}$ provably achieves a variance lower than or equal to that of the standard estimator $\hat{\mu}_{std}$ (i.e., $\mathrm{Var}(\hat{\mu}) \leq \mathrm{Var}(\hat{\mu}_{std})$).*

Substituting the estimators from Prop. 3.2 and Eq.(9) into Eq. (7) yields the Monte Carlo approximation of the BoN gradient:

$$\nabla_\theta J_{\mathrm{BoN}} \approx \sum_{i=1}^{K} p_i \left[ R(\mathbf{s}_{(i)}) \left( \nabla_\theta \ln \pi_\theta(\mathbf{s}_{(i)}) \right. \right.$$
$$\left. \left. + (N-1) \frac{1}{i-1} \sum_{j=1}^{i-1} \nabla_\theta \ln \pi_\theta(\mathbf{s}_{(j)}) \right) \right].$$
(12)

A notable property of this estimator is its intrinsic numerical stability. The combinatorial weights $p_i$ vanish for all $i < N$, which restricts the inner summation to cases where $i \geq N$, ensuring that the conditional gradient is always supported by at least $N-1$ samples and thus maintaining robust numerical stability.

Finally, by interchanging the order of summation in Eq. (12), we can restructure the nested terms into a linear weighted log-likelihood form, facilitating efficient computation:

$$\nabla_\theta J_{\mathrm{BoN}} \approx \sum_{i=1}^{K} \lambda_i \nabla_\theta \ln \pi_\theta(\mathbf{s}_{(i)}),$$
(13)

where the rank-dependent weight $\lambda_i$ is defined as:

$$\lambda_i = p_i R(\mathbf{s}_{(i)}) + (N-1) \sum_{j=i+1}^{K} \frac{p_j}{j-1} R(\mathbf{s}_{(j)}).$$
(14)

This formulation distinguishes our approach from standard reinforcement learning objectives $J_{std} = \sum \gamma_i \ln \pi_\theta(\mathbf{s}_{(i)})$, where weights typically depend solely on the scalar quality $R(\mathbf{s}_{(i)})$. In contrast, $\lambda_i$ incorporates rank-based interactions, modulating the gradient magnitude according to the sample's dual role: serving as a potential primary candidate and as a contrastive baseline within the Best-of-N selection process.

### 3.4. Practical Optimization via Rank-Based Policy Gradients

To efficiently optimize the Best-of-N objective with stable, low-variance updates, we incorporate group-based baseline statistics and a PPO-style clipping mechanism, formulating a practical algorithm within a GRPO-like framework.

**Variance Reduction via Baseline Subtraction.** The raw gradient estimator presented in Eq. (5) typically suffers from high variance. A standard mitigation strategy involves subtracting a baseline $b$—commonly the expected

return $b = \mathbb{E}_{\mathbf{s}^*}[R(\mathbf{s}^*)]$—yielding the modified formulation: $\mathbb{E}_{\mathbf{s}^*}[(R(\mathbf{s}^*) - b)\nabla_\theta \ln \pi_{bon}(\mathbf{s}^*)]$.

Let $\lambda_i'$ denote the weight corresponding to the shifted reward $R(\mathbf{s}) - b$ as defined in Eq. (14). While the update signal for each sample is constructed from a complex linear combination of rewards, this transformation preserves the unbiasedness of the estimator (i.e. $\mathbb{E}[\nabla J_{\mathrm{BoN}}(\lambda')] = \mathbb{E}[\nabla J_{\mathrm{BoN}}(\lambda)]$, proof provided in Appendix A.3). Here We estimated $b$ empirically over the batch as $\hat{b} = \sum_{k=1}^{K} p_k R(\mathbf{s}_{(k)})$.

**Proposition 3.3.** *(Proof provided in Appendix A.3) The weight $\lambda_i'$ resulting from subtracting the empirical Best-of-N expected return $\hat{b}$ is mathematically equivalent to the mean-centered original rank-based weight $\lambda_i$:*

$$\lambda_i' = \lambda_i - \bar{\lambda}, \quad where\ \bar{\lambda} = \frac{1}{K} \sum_{j=1}^{K} \lambda_j.$$
(15)

Proposition 3.3 establishes the theoretical justification for our optimization strategy: we can perform variance reduction efficiently without explicitly computing the baseline term during backpropagation as mean-centering the weights $\lambda_i$ implicitly executes the baseline subtraction.

**Surrogate Objective.** Guided by Proposition 3.3, we construct the standardized *Rank-based Advantage $A_i$* for each trajectory $\mathbf{s}_{(i)}$. We mean-center the weights to enforce the baseline subtraction and normalize by the batch standard deviation $\sigma_\lambda$ to ensure scale invariance, a prerequisite for stable trust-region optimization:

$$A_i = \frac{\lambda_i - \bar{\lambda}}{\sigma_\lambda},$$
(16)

where $\bar{\lambda}$ and $\sigma_\lambda$ denote the mean and standard deviation of weights within the batch, respectively.

To constrain the policy update, we adopt a PPO-style clipping mechanism. As this optimization paradigm necessitates per-step supervision, we broadcast the sequence-level scalar $A_i$ across all time steps $t$, effectively assigning uniform credit to every decision. Consequently, given the sampling policy $\pi_{\theta_{\mathrm{old}}}$ and a reference model $\pi_{\mathrm{ref}}$, the final surrogate objective adapts a GRPO-like formulation by substituting the standard return-based advantage with our derived rank-based advantage:

$$\mathcal{J}(\theta) = \mathbb{E}_{\mathbf{s} \sim \pi_{\theta_{\mathrm{old}}}} \left[ \frac{1}{K} \sum_{i=1}^{K} \frac{1}{T_i} \sum_{t=1}^{T_i} \min \left( \rho_{i,t}(\theta) A_i, \right. \right.$$
$$\left. \left. \mathrm{clip}(\rho_{i,t}(\theta), 1-\varepsilon, 1+\varepsilon) A_i \right) \right],$$
(17)

where $\rho_{i,t}(\theta) = \frac{\pi_\theta(a_{(i),t}|S_{(i),t})}{\pi_{\theta_{\mathrm{old}}}(a_{(i),t}|S_{(i),t})}$ denotes the probability ratio, $T_i$ represents the trajectory length, and $\varepsilon$ is the clipping

hyperparameter. We provide the complete pseudocode and a detailed analysis of the gradient dynamics derived from Eq. (17) in Appendix B.

## 4. Experiments

### 4.1. Experimental Setups

**Tasks and Datasets.** We evaluate FIDIA on two primary tasks.

**(1) High-Fidelity Motif Recovery.** Epitope and theozyme scaffolding demand strict preservation of functional geometries to ensure neutralizing and catalytic efficacy. Given generated backbones harboring functional motifs, the sequence design model is tasked to identify sequences that strictly maintain the native motif conformation upon folding, quantified by motif RMSD (mRMSD).

Benchmarking was performed on de novo backbones generated for the 24 standard benchmark motifs introduced in RFdiffusion (Watson et al., 2023). To assess the model's generalizability across different backbone distributions, we constructed two distinct test sets, **FrameF** and **RFD**, by generating 5 backbone scaffolds per motif (M=5) using FrameFlow (Yim et al., 2024) and RFdiffusion (Watson et al., 2023), respectively. To assess the model's applicability in realistic therapeutic design scenarios, we further validated it on an epitope scaffolding dataset for vaccine design comprising 11 targets (Correia et al., 2014; Castro et al., 2024; Watson et al., 2023), curated following the EVA benchmark (Huang et al., 2025).

**(2) Joint Optimization of Motif Recovery and Affinity.** Improving catalytic efficiency ($k_{cat}/K_m$) requires stabilizing the active site geometry while enhancing substrate capture. Consequently, the model is tasked with identifying sequences that minimize mRMSD while concurrently optimizing Vina scores for the enzyme-substrate complex. We derived a dataset from the BioLiP database (Yang et al., 2013), where 54 enzyme-ligand complexes with annotated catalytic residues were specifically set aside for evaluation. Specific implementation details and training configurations of these two tasks are provided in Appendix C.

**Sampling and Metrics.** We employ a BoN (N=8) sampling protocol, generating eight candidate sequences for each target backbone to report optimal mRMSD (mRMSD*), global RMSD (RMSD*), and TM-score (TM-score*). A backbone is designated as successful if at least one generated sequence satisfies the dual criteria of mRMSD < 1 and RMSD < 2. We report the Success Rate (SR) of target backbones and the total count of successfully recovered motifs (Solved) to evaluate the robustness of the entire design pipeline.

**Baselines.** We benchmark our method against state-of-the-art inverse folding models, including ProteinMPNN (Dauparas et al., 2022), ESM-IF (Hsu et al., 2022), and LigandMPNN (Dauparas et al., 2025). For reinforcement learning (RL) fine-tuning, we compare our method with widely adopted baselines, specifically GRPO (Wang et al., 2025) and multi-round DPO (Xu et al., 2025). We also benchmark against algorithms derived from the BoN strategy. These baselines involve training on the top-1 or top-$k$ candidates from a sampled batch, referred to as $BoN_{top1}$ and $BoN_{topk}$ (Touvron et al., 2023). To ensure a fair comparison, our method and all training-based baselines (including both RL and BoN variants) are built upon the same pre-trained ProteinMPNN backbone.

### 4.2. Performance on General Benchmarks

As summarized in Table 1, FIDIA achieves superior performance across primary metrics on both datasets. Specifically, on the FrameF dataset, FIDIA increases the Success Rate (SR) from 26.67% (MPNN) to 37.50%, surpassing the strongest baselines (MultiDPO and $BoN_{top1}$, 34.17%). This superior performance extends to the RFD dataset, where FIDIA achieves a remarkable SR of 65.83%. In terms of structural accuracy, FIDIA further reduces the mRMSD* to 1.43 Å (FrameF) and 0.94 Å (RFD), outperforming the competitive BoN baselines. Moreover, FIDIA achieves these accuracy gains while simultaneously demonstrating superior structural confidence (pLDDT) and sample diversity (Diversity). This dual-source validation ensures that our sequence design evaluation is robust and not biased toward a specific structural generator.

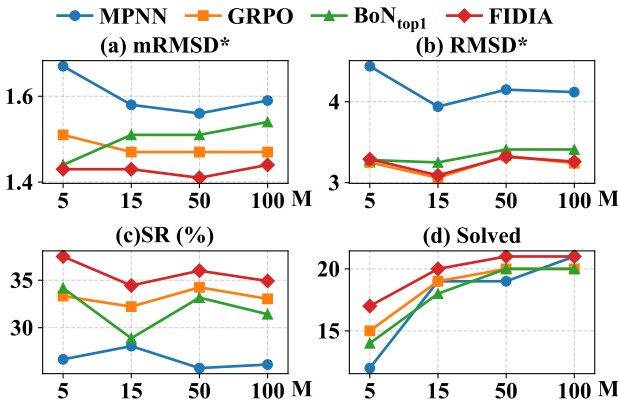

*Figure 1.* **Evaluation of the complete design pipeline across varying structure sampling budgets (M).**

To provide an independent assessment of structural fidelity, we cross-validate our generated sequences using AlphaFold2 (AF2) (Jumper et al., 2021). The results in Table 2 reveal a high consistency between the two structure

*Table 1.* Performance comparison on FrameF and RFD datasets. Best results are **bolded** and second best are underlined.

| Method | SR(%) ↑ | | mRMSD* ↓ | | RMSD* ↓ | | TM-score* ↑ | | mpLDDT* ↑ | | pLDDT* ↑ | | Diversity ↑ | |
|---|---|---|---|---|---|---|---|---|---|---|---|---|---|---|
| | FrameF | RFD | FrameF | RFD | FrameF | RFD | FrameF | RFD | FrameF | RFD | FrameF | RFD | FrameF | RFD |
| *Base Models* | | | | | | | | | | | | | | |
| MPNN | 26.67 | 55.00 | 1.67 | 1.25 | 4.44 | 2.26 | 0.69 | 0.85 | 69.12 | **82.73** | 69.92 | 84.34 | 0.26 | 0.31 |
| ESM-IF | 15.83 | 34.17 | 1.84 | 1.64 | 5.05 | 3.76 | 0.64 | 0.76 | 67.76 | 76.77 | 68.43 | 78.73 | 0.14 | 0.16 |
| *RL / BoN Baselines* | | | | | | | | | | | | | | |
| MultiDPO | 34.17 | 57.50 | 1.50 | 1.17 | 3.53 | 1.98 | 0.73 | 0.85 | 70.93 | 82.50 | **72.52** | **84.48** | 0.22 | 0.28 |
| GRPO | 33.33 | 58.33 | 1.51 | 1.14 | **3.25** | 1.88 | **0.74** | 0.85 | 70.13 | 81.53 | 71.86 | 83.23 | 0.23 | 0.29 |
| BoN$_\text{top1}$ | 34.17 | 55.00 | 1.44 | 1.10 | 3.28 | 1.70 | 0.73 | 0.85 | 69.19 | 80.67 | 70.46 | 82.48 | 0.26 | 0.31 |
| BoN$_\text{topk}$ | 28.33 | 53.33 | 1.61 | 1.06 | 3.38 | 1.91 | 0.73 | 0.84 | 68.92 | 81.04 | 70.50 | 82.78 | 0.22 | 0.24 |
| *Ours* | | | | | | | | | | | | | | |
| **FIDIA** | **37.50** | **65.83** | **1.43** | **0.94** | 3.29 | **1.59** | **0.74** | **0.87** | **71.05** | 82.72 | 72.12 | 84.18 | **0.35** | **0.43** |

*Table 2.* Cross-validation of design performance using AlphaFold2 (AF2) to verify robustness beyond the training oracle (ESMFold).

| Method | SR(%) ↑ | | mRMSD* ↓ | | RMSD* ↓ | | TM-score* ↑ | | mpLDDT* ↑ | | pLDDT* ↑ | |
|---|---|---|---|---|---|---|---|---|---|---|---|---|
| | ESMFold | AF2 | ESMFold | AF2 | ESMFold | AF2 | ESMFold | AF2 | ESMFold | AF2 | ESMFold | AF2 |
| MPNN | 26.67 | 30.00 | 1.67 | 1.58 | 4.44 | 4.38 | 0.69 | 0.71 | 69.12 | 78.46 | 69.92 | 77.90 |
| GRPO | 33.33 | 31.67 | 1.51 | 1.34 | **3.25** | 3.04 | **0.74** | 0.76 | 70.13 | 78.09 | 71.86 | 78.87 |
| BoN$_\text{top1}$ | 34.17 | 34.17 | 1.44 | **1.24** | 3.28 | 3.20 | 0.73 | 0.75 | 69.19 | 77.01 | 70.46 | 76.69 |
| **FIDIA** | **37.50** | **36.67** | **1.43** | 1.29 | 3.29 | **2.86** | **0.74** | **0.77** | **71.05** | **80.25** | **72.12** | **80.18** |

prediction methods, confirming that our model's improvements reflect actual gains in design quality. Notably, FIDIA maintains its superiority under AF2 evaluation, achieving the highest Success Rate (36.67%) and structural metrics (1.29Å mRMSD, 80.18 pLDDT*).

We evaluate the complete pipeline across varying structure sampling budgets (M), where performance is primarily quantified by the number of successfully recovered motifs (**Solved**). As shown in Figure 1, FIDIA consistently outperforms baselines across all sampling scales, maintaining the lowest mRMSD* and the highest number of solved targets. Notably, our method converges to optimal performance at M=50, recovering the maximum number of targets (21) and achieving high structural fidelity. This indicates that FIDIA alleviates the need for extensive structural sampling per motif through its superior sequence sampling capability. To further validate the efficiency of FIDIA, a complementary evaluation across varying sequence sampling budgets ($N$) is detailed in Appendix D.4.

### 4.3. Real-world Design Capabilities

**High-Precision Vaccine Scaffolding.** To demonstrate FIDIA's practical utility in therapeutic design, we applied it to design sequences for viral epitope scaffolds generated by RFdiffusion. As detailed in Table 3, FIDIA outperforms all baselines, achieving the highest Success Rate (27.09%) and accurately solving the most targets (7/11). These results confirm the effectiveness of FIDIA in high-precision

structural engineering.

*Table 3.* Performance comparison on the epitope scaffolding dataset.

| Method | SR | mRMSD* | RMSD* | mpLDDT* | Solved |
|---|---|---|---|---|---|
| MPNN | 25.45 | 3.06 | 4.17 | 63.42 | 6 |
| GRPO | 25.64 | 2.85 | 3.63 | 64.14 | 5 |
| BoN$_\text{top1}$ | 24.91 | 2.96 | 3.77 | 63.13 | 5 |
| **FIDIA** | **27.09** | **2.65** | **3.31** | **65.65** | **7** |

Figure 2 presents a representative case study on the sequence design for a RFdiffusion-generated backbone. Specifically, this backbone is designed to scaffold the Respiratory Syncytial Virus (RSV) Fusion Glycoprotein Site 0 epitope, which consists of two discontinuous segments(derived from PDB: `4JHW`, Chain F). While baseline methods exhibit noticeable deviations in the motif region, the structure of the FIDIA-designed sequence closely respects the motif geometry, achieving an mRMSD of 0.79 Åand a global RMSD of 0.93 Å.

**Affinity-Enhancing Enzyme Design.** In this task, our objective is to maximize ligand binding affinity while preserving the structural integrity of designed sequences. Specifically, we fine-tuned LigandMPNN via FIDIA, utilizing AutoDock Vina scores as functional rewards to guide the policy toward high-affinity sequence space. As shown in Table 4, FIDIA achieves a superior binding affinity with a Vina score of -7.21 kcal/mol, outperforming both the standard

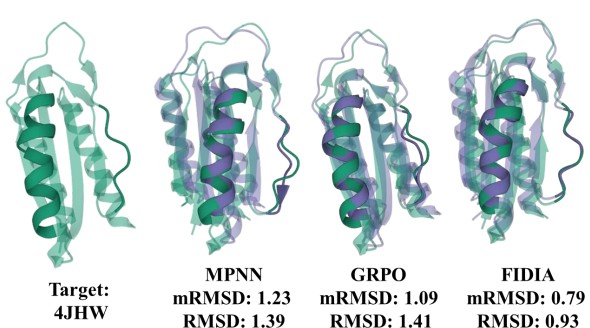

**Target:**
**4JHW**

**MPNN**
**mRMSD: 1.23**
**RMSD: 1.39**

**GRPO**
**mRMSD: 1.09**
**RMSD: 1.41**

**FIDIA**
**mRMSD: 0.79**
**RMSD: 0.93**

*Figure 2.* **Case study on an RFdiffusion-generated scaffold harboring the RSVF0.** The ground truth target structure is colored in green, and the structures refolded from the designed sequences are shown in purple. The discontinuous motif regions are highlighted in solid representation, contrasting with the transparent scaffold structure.

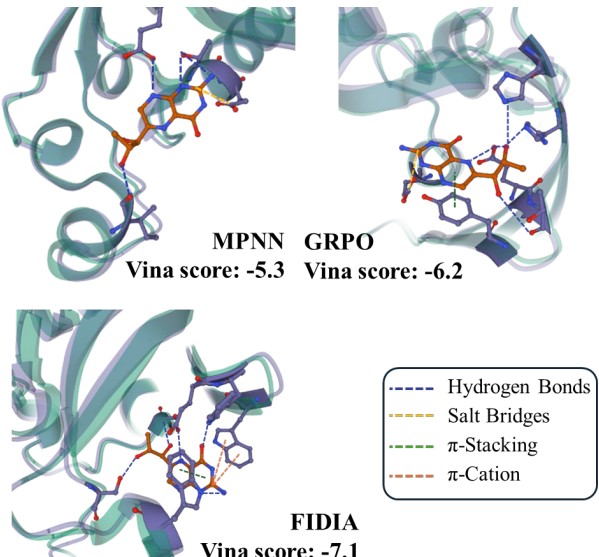

*Figure 3.* **Case study on an enzyme-substrate complex.** (PDB `1B66`). The target structure (green) and the refolded structures of the designed sequences (purple) are superimposed. FIDIA achieves improved binding affinity by establishing a dense network of noncovalent interactions compared to the sparser contacts observed in MPNN and GRPO baselines.

MPNN and the GRPO baseline trained with the identical reward signal, while maintaining competitive structural fidelity.

Figure 3 presents a case study of 6-pyruvoyltetrahydropterin synthase (PDB `1b66`) complexed with the ligand BIO. While all methods preserve the target backbone fold, FIDIA effectively optimizes the physicochemical environment of the binding pocket. PLIP analysis   (Adasme et al., 2021) indicates that structure of FIDIA-designed sequence establishes a robust interaction network—comprising six hydrogen bonds, one pi-stacking, and two pi-cation interactions—whereas baselines form fewer stabilizing contacts, resulting in weaker binding modes.

*Table 4.* Performance comparison on the affinity-enhancing enzyme design dataset

| Method | Vina | SR | mRMSD* | RMSD* | Diversity |
|---|---|---|---|---|---|
| LigandMPNN | -6.53 | **90.74** | **0.56** | **1.13** | **0.18** |
| GRPO | -7.06 | 75.93 | 0.66 | 1.39 | 0.13 |
| **FIDIA** | **-7.21** | 83.33 | 0.64 | 1.29 | 0.17 |

## 5. Discussion

In this paper, we introduce FIDIA, a framework that empowers inverse folding models to satisfy complex biological constraints under the Best-of-N inference protocol. Empirically, FIDIA demonstrates superior performance in both motif scaffolding and affinity-enhancing enzyme design, validating its generalization capability across diverse biochemical landscapes. However, several limitations remain to be addressed: (i) the computational overhead induced by iterative batch sampling; (ii) the reliance on the fidelity of surrogate reward oracles; and (iii) the necessity for retraining when adapting to novel tasks or base architectures. The

proposed method for aligning the Best-of-N (BoN) inference protocol with training objectives can also be extended to other domains that rely on BoN, such as small molecule design, mathematical reasoning, and code generation.

## Impact Statement

Our work on function-informed sequence design, FIDIA, can be utilized in the sequence design stage of specific protein engineering tasks such as vaccine development and enzyme scaffolding. It is also essential to ensure the responsible use of our method and refrain from using it for any harmful biological or computational purposes.

## Code Availability

Our code and datasets are publicly released at `https://github.com/deng-ai-lab/FIDIA-code` and `https://zenodo.org/records/20353469`.

## Acknowledgements

This work was supported by the National Natural Science Foundation of China under Grant 62325101 and Zhongguancun Academy under Grant 02012503.

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

# A. Proof of results

## A.1. Proof of Prop.3.1

*Proof.* By definition, the cumulative distribution function is given by the integral over the trajectory space $\Omega$:

$$F_\theta(r) = \int_\Omega \mathbb{I}(R(\mathbf{s}) \leq r)\pi_\theta(\mathbf{s})\,\mathrm{d}\mathbf{s}. \tag{18}$$

Taking the gradient with respect to $\theta$, and assuming the validity of interchanging the gradient and integral operations, we have:

$$\begin{aligned} \nabla_\theta F_\theta(r) &= \nabla_\theta \int_\Omega \mathbb{I}(R(\mathbf{s}) \leq r)\pi_\theta(\mathbf{s})\,\mathrm{d}\mathbf{s} \\ &= \int_\Omega \mathbb{I}(R(\mathbf{s}) \leq r)\nabla_\theta \pi_\theta(\mathbf{s})\,\mathrm{d}\mathbf{s}. \end{aligned} \tag{19}$$

Using the log-derivative trick, $\nabla_\theta \pi_\theta(\mathbf{s}) = \pi_\theta(\mathbf{s})\nabla_\theta \ln \pi_\theta(\mathbf{s})$, we substitute this into the integral:

$$\nabla_\theta F_\theta(r) = \int_\Omega \mathbb{I}(R(\mathbf{s}) \leq r)\pi_\theta(\mathbf{s})\nabla_\theta \ln \pi_\theta(\mathbf{s})\,\mathrm{d}\mathbf{s}. \tag{20}$$

Now, considering the gradient of the log-CDF:

$$\nabla_\theta \ln F_\theta(r) = \frac{\nabla_\theta F_\theta(r)}{F_\theta(r)} = \int_\Omega \frac{\mathbb{I}(R(\mathbf{s}) \leq r)\pi_\theta(\mathbf{s})}{F_\theta(r)}\nabla_\theta \ln \pi_\theta(\mathbf{s})\,\mathrm{d}\mathbf{s}. \tag{21}$$

We observe that the term $\frac{\mathbb{I}(R(\mathbf{s}) \leq r)\pi_\theta(\mathbf{s})}{F_\theta(r)}$ corresponds exactly to the conditional probability density function of $\mathbf{s}$ given $R(\mathbf{s}) \leq r$:

$$\pi_\theta(\mathbf{s} \mid R(\mathbf{s}) \leq r) = \frac{\mathbb{I}(R(\mathbf{s}) \leq r)\pi_\theta(\mathbf{s})}{P(R(\mathbf{s}) \leq r)}. \tag{22}$$

Substituting this back, we obtain the expectation form:

$$\nabla_\theta \ln F_\theta(r) = \mathbb{E}_{\mathbf{s} \sim \pi_\theta(\cdot|R(\mathbf{s}) \leq r)}\left[\nabla_\theta \ln \pi_\theta(\mathbf{s})\right]. \tag{23}$$

$\square$

*Remark* A.1. Although Eq. (5) resembles the standard score function expectation $\mathbb{E}_{\mathbf{s} \sim \pi_\theta}[\nabla_\theta \ln \pi_\theta(\mathbf{s})]$, which typically evaluates to zero, the expectation here is taken over a **restricted domain** (the sub-level set). Consequently, this term is generally non-zero, representing the sensitivity of the probability mass within the region $R(\mathbf{s}) \leq r$ to changes in parameters $\theta$.

## A.2. Proof of Prop.3.2

### A.2.1. PROOF OF UNBIASEDNESS

*Proof.* We construct a U-statistic over the batch $\mathcal{B}$. Let $\mathcal{S}$ denote the collection of all possible subsets of $\mathcal{B}$ with size $N$. The total number of such subsets is $|\mathcal{S}| = \binom{K}{N}$. For any subset $S \in \mathcal{S}$, let $\mathbf{s}_{max}(S)$ be the trajectory in $S$ with the maximum reward. Since all trajectories in $\mathcal{B}$ are i.i.d. samples from $\pi_\theta$, any random subset $S$ is distributionally equivalent to sampling $N$ trajectories directly. Thus:

$$\mathbb{E}\left[f(\mathbf{s}_{max}(S))\right] = \mathbb{E}[f(\mathbf{s}^*)] = \mu. \tag{24}$$

We define the U-statistic as the average over all subsets:

$$U = \frac{1}{\binom{K}{N}} \sum_{S \in \mathcal{S}} f(\mathbf{s}_{max}(S)). \tag{25}$$

By linearity of expectation, $\mathbb{E}[U] = \mu$. To compute $U$ efficiently without enumerating all subsets, we group the terms by the sorted trajectories $\mathbf{s}_{(i)}$. For a specific sorted trajectory $\mathbf{s}_{(i)}$ to be the maximum $\mathbf{s}_{max}(S)$ of a subset $S$, two conditions must

be met: 1. $\mathbf{s}_{(i)}$ must be included in $S$. 2. The remaining $N-1$ trajectories in $S$ must have rewards less than or equal to $R(\mathbf{s}_{(i)})$. In our sorted batch $\mathcal{B}$ (where $R(\mathbf{s}_{(1)}) \leq \cdots \leq R(\mathbf{s}_{(K)})$), the trajectories satisfying the second condition are exactly those with indices $\{1, \ldots, i-1\}$. The number of ways to choose $N-1$ trajectories from these $i-1$ candidates is $\binom{i-1}{N-1}$. Thus, in the summation over $\mathcal{S}$, the term $f(\mathbf{s}_{(i)})$ appears exactly $\binom{i-1}{N-1}$ times. We can rewrite $U$ as:

$$U = \sum_{i=1}^{K} \frac{\binom{i-1}{N-1}}{\binom{K}{N}} f(\mathbf{s}_{(i)}). \tag{26}$$

This matches the definition of $\hat{\mu}$, proving that $\hat{\mu}$ is unbiased. $\qquad\square$

### A.2.2. PROOF OF VARIANCE REDUCTION

*Proof.* We compare the proposed estimator $\hat{\mu}$ against the standard Monte Carlo estimator under the same total computational budget $K$.

The standard ("naive") approach to estimate $\mu$ with budget $K$ is to generate $M = \lfloor K/N \rfloor$ independent groups of samples, where each group $j$ consists of $N$ trajectories. Let $Y_j$ be the result computed from the $j$-th group. The naive estimator is the average:

$$\hat{\mu}_{naive} = \frac{1}{M} \sum_{j=1}^{M} Y_j. \tag{27}$$

Since all trajectories are sampled i.i.d. from $\pi_\theta$, the following two data generation processes are stochastically equivalent:

- Generate $M$ groups of $N$ samples one by one.

- Generate a single pooled batch $\mathcal{B}$ of $K$ samples, and then randomly partition $\mathcal{B}$ into $M$ disjoint groups $G_1, \ldots, G_M$ of size $N$.(due to the i.i.d. , a fix order can be seen as a randomly partition)

We analyze the variance using the perspective of latter Process. This allows us to condition on the realized set of values in the batch $\mathcal{B}$.

If we fix the batch $\mathcal{B}$ and average $\hat{\mu}_{naive}$ over all possible ways to partition $\mathcal{B}$ into $M$ groups, we recover the average over all size-$N$ subsets, which is exactly our estimator:

$$\mathbb{E}[\hat{\mu}_{naive} \mid \mathcal{B}] = \hat{\mu}. \tag{28}$$

By the Law of Total Variance:

$$\begin{aligned} \text{Var}(\hat{\mu}_{naive}) &= \text{Var}(\mathbb{E}[\hat{\mu}_{naive} \mid \mathcal{B}]) + \mathbb{E}[\text{Var}(\hat{\mu}_{naive} \mid \mathcal{B})] \\ &= \text{Var}(\hat{\mu}) + \mathbb{E}[\text{Var}(\hat{\mu}_{naive} \mid \mathcal{B})]. \end{aligned} \tag{29}$$

Since the conditional variance $\text{Var}(\hat{\mu}_{naive} \mid \mathcal{B})$ represents the noise introduced by random partitioning and is always non-negative, we conclude:

$$\text{Var}(\hat{\mu}) \leq \text{Var}(\hat{\mu}_{naive}). \tag{30}$$

This proves that $\hat{\mu}$ is strictly more efficient (or equal) for any non-trivial distribution of rewards.Intuitively, this reflects the essence of Rao-Blackwellization: our estimator eliminates the stochastic noise associated with arbitrary subset partitioning, thereby retaining only the variance inherent to the data distribution. $\qquad\square$

### A.3. Proof of Prop. 3.3

To prove that centering the weights is equivalent to subtracting the expected Best-of-N return as a baseline, we establish two lemmas regarding the properties of the rank-based weights $\lambda_i$. Recall the definition: $\lambda_i = p_i R(\mathbf{s}_{(i)}) + (N - 1) \sum_{j=i+1}^{K} \frac{p_j}{j-1} R(\mathbf{s}_{(j)})$, with $p_k = \binom{k-1}{N-1}/\binom{K}{N}$.

**Lemma A.2.** *The geometric term associated with a constant baseline shift is uniform across all $i$:*

$$C_i \triangleq p_i + (N-1) \sum_{j=i+1}^{K} \frac{p_j}{j-1} = \frac{N}{K}. \tag{31}$$

*Proof.* Substituting $p_k$ into $C_i$ and applying the identity $\frac{k}{n}\binom{n}{k} = \binom{n-1}{k-1}$ to the summation term yields:

$$C_i = \frac{1}{\binom{K}{N}} \left[ \binom{i-1}{N-1} + \sum_{j=i+1}^{K} \binom{j-2}{N-2} \right]. \tag{32}$$

The term in the brackets forms a summation sequence: $\binom{i-1}{N-1} + \binom{i-1}{N-2} + \cdots + \binom{K-2}{N-2}$. By iteratively applying Pascal's Identity, $\binom{n}{k} + \binom{n}{k-1} = \binom{n+1}{k}$ (the Hockey-stick identity), the sum collapses to $\binom{K-1}{N-1}$. Thus:

$$C_i = \frac{\binom{K-1}{N-1}}{\binom{K}{N}} = \frac{N!(K-N)!(K-1)!}{(N-1)!(K-N)!K!} = \frac{N}{K}. \tag{33}$$

$\square$

**Lemma A.3.** *The sum of the rank-based weights corresponds to the scaled empirical BoN objective:*

$$\sum_{i=1}^{K} \lambda_i = N \sum_{k=1}^{K} p_k R(\mathbf{s}_{(k)}) = N b_{BoN}. \tag{34}$$

*Proof.* We derive this by inspecting the coefficient sum in Eq. 12. By setting the gradient vectors $\nabla_\theta \ln \pi_\theta(\mathbf{s}_{(k)}) = 1$ for all $k$, the identity simplifies as follows:

**RHS:** The linear combination becomes $\sum_{i=1}^{K} \lambda_i$.

**LHS:** The inner term becomes $1 + (N-1)\frac{1}{i-1}(i-1) = N$. The full expression simplifies to $\sum_{i=1}^{K} p_i R(\mathbf{s}_{(i)}) \cdot N = N b_{\text{BoN}}$.

Equating LHS and RHS concludes the proof. $\square$

**Proof** Consider a modified reward function with a constant baseline $R'(\mathbf{s}) = R(\mathbf{s}) - b$. Due to the linearity of $\lambda_i$ with respect to $R$, and using Lemma A.2, the transformed weight becomes:

$$\begin{aligned}
\lambda_i' &= p_i(R(\mathbf{s}_{(i)}) - b) + (N-1) \sum_{j=i+1}^{K} \frac{p_j}{j-1}(R(\mathbf{s}_{(j)}) - b) \\
&= \lambda_i - b \cdot C_i \\
&= \lambda_i - b\frac{N}{K}.
\end{aligned} \tag{35}$$

Substituting $\lambda_i'$ into the gradient estimator:

$$\nabla_\theta J' = \sum_{i=1}^{K} \lambda_i' \nabla_\theta \ln \pi_\theta(\mathbf{s}_{(i)}) = \underbrace{\sum_{i=1}^{K} \lambda_i \nabla_\theta \ln \pi_\theta(\mathbf{s}_{(i)})}_{\text{Original Gradient}} - \underbrace{b\frac{N}{K} \sum_{i=1}^{K} \nabla_\theta \ln \pi_\theta(\mathbf{s}_{(i)})}_{\text{Baseline Term}}. \tag{36}$$

Since the expected score function is zero, i.e., $\mathbb{E}_{\mathbf{s}\sim\pi_\theta}[\nabla_\theta \ln \pi_\theta(\mathbf{s})] = 0$, the expected value of the Baseline Term vanishes. Thus, $\mathbb{E}[\nabla_\theta J'] = \mathbb{E}[\nabla_\theta J]$, proving that the estimator remains unbiased.

Finally, we choose the baseline to be the empirical expected return, $b = b_{\text{BoN}}$. From Lemma A.3, we have $N b_{\text{BoN}} = \sum_{j=1}^{K} \lambda_j$. The shift term becomes:

$$b\frac{N}{K} = b_{\text{BoN}}\frac{N}{K} = \frac{1}{K} \sum_{j=1}^{K} \lambda_j = \bar{\lambda}. \tag{37}$$

Therefore, $\lambda_i' = \lambda_i - \bar{\lambda}$, confirming that subtracting the expected BoN return is mathematically equivalent to centering the weights. $\qquad\square$

## B. Algorithmic Details

### B.1. Gradient Dynamics

Substituting the standard advantage with our rank-based advantage $A_i$ fundamentally reshapes the optimization landscape through two distinct mechanisms:(1) **Uniform Suppression of Non-Competitive Samples** ($i < N$)**:** For trajectories ranked below $N$, the probability $p_i$ of being the maximum vanishes. Consequently, the advantage $A_i$ for these samples becomes uniformly negative. This effectively treats the lower tail of the distribution as a homogeneous contrastive background, suppressing these sequences indiscriminately rather than distinguishing between varying degrees of sub-optimality. (2) **Rank-Dependent Amplification of Top-Tier Samples** ($i \geq N$)**:** For the highest-ranking candidates, the weight $\lambda_i$ exhibits super-linear growth driven by the combinatorial factor $p_i$. This yields a sharpened gradient signal that disproportionately incentivizes improvements in the highest-performing subset. Unlike standard RL, which optimizes the expected return, this mechanism focuses on the upper tail of the distribution, effectively steering the search toward high-fitness regimes within the sequence landscape.

### B.2. Pseudocode

---

**Algorithm 1** FIDIA: Function-Informed Sequence Design via Inference-Aligned Policy Optimization

---

1: **Input:** Dataset of backbone-condition pairs $\mathcal{D} = \{(\mathbf{x}, c)\}$, Pretrained Policy $\pi_\theta$, Batch size $K$, Inference budget $N$, Clipping $\varepsilon$.
2: **while** not converged **do**
3:      Sample a batch of pairs $\{(\mathbf{x}, c)\}$ from $\mathcal{D}$
4:      $\pi_{\theta_{old}} \leftarrow \pi_\theta$
5:      **for** each pair $(\mathbf{x}, c)$ in batch **do**
6:          *// 1. Batch Sampling conditioned on functional constraints*
7:          Sample $K$ sequences $\{\mathbf{s}_1, \ldots, \mathbf{s}_K\} \sim \pi_{\theta_{old}}(\cdot|\mathbf{x}, c)$
8:          Compute rewards $\mathbf{r} = \{R(\mathbf{s}_k, \mathbf{x}, c)\}_{k=1}^K$ via Oracles
9:          *// 2. Rank-Based Weight Calculation*
10:         Sort sequences: $\mathbf{s}_{(1)}, \ldots, \mathbf{s}_{(K)}$ such that $R(\mathbf{s}_{(1)}) \leq \cdots \leq R(\mathbf{s}_{(K)})$
11:         Compute combinatorial probabilities $p_i = \binom{i-1}{N-1}/\binom{K}{N}$ for $i \in [1, K]$
12:         **for** $i = 1$ **to** $K$ **do**
13:            Compute rank-based weight $\lambda_i$ (Eq. 13):
14:            $\lambda_i \leftarrow p_i R(\mathbf{s}_{(i)}) + (N-1) \sum_{j=i+1}^K \frac{p_j}{j-1} R(\mathbf{s}_{(j)})$
15:         **end for**
16:         *// 3. Advantage Estimation via Baseline Subtraction*
17:         Compute mean $\bar{\lambda} = \frac{1}{K} \sum \lambda_i$ and std $\sigma_\lambda$
18:         Compute standardized rank-based advantage for each sequence:
19:         $A_i \leftarrow \frac{\lambda_i - \bar{\lambda}}{\sigma_\lambda}$
20:      **end for**
21:      *// 4. Policy Optimization*
22:      Compute loss $\mathcal{J}(\theta)$ on the batch:
23:      $\mathcal{L} = \frac{1}{|\mathcal{B}|K} \sum_{(\mathbf{x}, c)} \sum_{i=1}^K \left[\min\left(\rho_i A_i, \mathrm{clip}(\rho_i, 1-\varepsilon, 1+\varepsilon)A_i\right)\right]$
24:      Update $\theta$ by maximizing $\mathcal{L}$
25: **end while**

---

# C. Implementation details

## C.1. Datasets

We trained our models on two distinct datasets tailored to specific tasks. For synthetic scaffolding, we utilized the CATH S20 non-redundant dataset, filtering for chains with a length of fewer than 150 residues. This resulted in 2,881 backbones. During training, each backbone was paired with 1 to 3 pre-sampled disjoint motif segments covering 5%–50% of the total sequence length. For the enzyme scaffolding task, we curated a dataset from BioLip (Yang et al., 2013). To ensure high-fidelity docking with AutoDock Vina, we filtered ligand-protein complexes based on physicochemical properties—specifically molecular weight, rotatable bonds, and the absence of metal coordination. Protein lengths were similarly restricted to 150 residues. We reserved complexes with annotated catalytic residues for the test set ($n = 54$) and used the remainder for training ($n = 1,615$). Motifs in this setting were defined to include both binding and catalytic residues.

## C.2. Training Configuration.

We employed ProteinMPNN  (Dauparas et al., 2022) and LigandMPNN  (Dauparas et al., 2025) as the foundational policies for the vaccine and enzyme scaffolding tasks, respectively. We set the trajectory parameters to $K = 16$ and $N = 8$, balancing computational efficiency with sufficient exploration of the sequence space. Optimization was performed using the Adam optimizer with a batch size of 32, 8 gradient accumulation steps, and a cosine learning rate schedule with warmup peaking at $5 \times 10^{-5}$. A clip range of 1.0 was applied. For our proposed method, FIDIA, we set the training temperature to 0.3 to mitigate sensitivity to discrepancies between training and evaluation policies. In contrast, baseline methods utilized a sampling temperature of 1.0, following established protocols  (Wang et al., 2025; Xu et al., 2025). Ablation studies regarding the training temperature setting are provided in Appendix D.3. All experiments were conducted on NVIDIA A800 GPUs.

## C.3. Evaluation Configuration

To evaluate the proposed sequence design methods, we adopt a **Best-of-$N$ (BoN)** inference protocol consisting of three stages: (i) generating $N = 8$ sequences for each designed backbone; (ii) refolding these sequences using a state-of-the-art structure-prediction model; and (iii) selecting the candidate with the minimum **Motif RMSD** as the final design.

Consistent with this protocol, we report the **minimum Motif RMSD (mRMSD*)**, representing the geometric deviation of the best-performing sequence. For a comprehensive structural assessment, we also report the **minimum global RMSD (RMSD*)** and the **maximum TM-score (TM*)**. The **Success Rate (SR)** is defined as the percentage of backbones for which at least one generated sequence satisfies the dual criteria of mRMSD $< 1.0\,\text{Å}$ and global RMSD $< 2.0\,\text{Å}$.

For the **Vaccine Scaffolding** task, we evaluate the end-to-end pipeline using two aggregate metrics: the **Motif Success Count**, which denotes the number of targets with at least one successful design (as defined above), and the **Median of Minimum mRMSD**, which reflects the overall precision across all motifs. For **Enzyme Design**, we report the **minimum Vina Score** and the **minimum Motif RMSD** achieved for each redesigned enzyme-substrate complex to quantify binding affinity and catalytic site stability, respectively.

## C.4. Model Architecture and Input Features

To explicitly inform the model of the functional constraints, we augment the standard node features with a binary *task-aware mask*. Specifically, for each residue node $i$, we append a binary indicator $f_i \in \{0, 1\}$, where $f_i = 1$ if the residue belongs to the fixed functional motif and $f_i = 0$ for the redesignable scaffold regions. This allows the encoder to distinguish between constrained and flexible nodes during message passing.

## C.5. Reward Design

To address the multi-objective nature of these scenarios, we design a composite reward function $R(\mathbf{s})$ that flexibly integrates structural and functional metrics. The total reward is defined as:

$$R(\mathbf{s}) = w_{\text{motif}}r_{\text{motif}} + w_{\text{global}}r_{\text{global}} + w_{\text{vina}}r_{\text{vina}} + w_{\text{div}}r_{\text{div}} \tag{38}$$

The specific components are formulated as follows:

**1. Motif Consistency ($r_{\textbf{motif}}$):** To encourage precise recovery of the functional site, we employ a Cauchy kernel on the

Motif RMSD. Its heavy-tailed distribution prevents gradient vanishing for samples with initial high deviation:

$$r_{\text{motif}} = \frac{1}{1 + (\text{RMSD}_{\text{motif}}/\tau)^2}, \quad \tau = 1.0\text{Å} \tag{39}$$

**2. Global Structural Constraint ($r_{\textbf{global}}$):** Unlike the motif, the scaffold backbone requires satisfaction of a constraint rather than infinite minimization. We propose a *Flat-top Cauchy* function that assigns a full reward of 1.0 if the RMSD is within a tolerance threshold $\delta$, tolerating minor fluctuations:

$$r_{\text{global}} = \begin{cases} 1.0 & \text{if RMSD}_{\text{global}} \leq \delta \\ \frac{1}{1+((\text{RMSD}_{\text{global}}-\delta)/\tau)^2} & \text{if RMSD}_{\text{global}} > \delta \end{cases} \tag{40}$$

where $\delta = 2.0\text{Å}$ and $\tau = 2.0\text{Å}$.

**3. Binding Affinity ($r_{\textbf{vina}}$):** For the enzyme task, we quantify functionality using AutoDock Vina energies $E_{\text{vina}}$. We map these unbounded values to a $[0, 1]$ range using a linear transformation bounded by a baseline energy threshold $E_{\text{baseline}}$ and a target affinity $E_{\text{target}}$:

$$r_{\text{vina}} = \text{clip}\left(\frac{E_{\text{baseline}} - E_{\text{vina}}}{E_{\text{baseline}} - E_{\text{target}}}, 0.0, 1.0\right) \tag{41}$$

We set $E_{\text{baseline}} = -2.0$ kcal/mol and $E_{\text{target}} = -12.0$ kcal/mol.

**4. Sequence Diversity ($r_{\textbf{div}}$):** To mitigate mode collapse and encourage the model to explore distinct solutions, we calculate a diversity score within the sampled batch $\mathcal{S} = \{s_1, \ldots, s_N\}$ for each backbone. For a sequence $s_i$, the reward is the average normalized Hamming distance to all other candidates in the batch:

$$r_{\text{div}}(s_i) = \frac{1}{N-1} \sum_{j \neq i, j \in \mathcal{S}} \mathcal{H}(s_i, s_j) \tag{42}$$

where $\mathcal{H}(s_i, s_j) \in [0, 1]$ denotes the proportion of mismatched amino acids between sequence $s_i$ and $s_j$. A higher $r_{\text{div}}$ indicates that $s_i$ is unique compared to its peers.

The weights are adjusted per task: for Vaccine Scaffolding, we focus on structure ($w_{\text{motif}} = 4.0, w_{\text{global}} = 4.0, w_{\text{vina}} = 0.0, w_{\text{div}} = 1.0,$); for Enzyme Scaffolding, we incorporate affinity optimization ($w_{\text{motif}} = 3.0, w_{\text{global}} = 4.0, w_{\text{vina}} = 14.0, w_{\text{div}} = 1.0,$). Our scalar weights are derived from the distinct optimization dynamics of each task. For enzyme scaffolding, optimizing Vina affinity is much harder than improving structural metrics. Thus, we set the Vina weight (14) to twice the combined structural weights (3+4) to balance the reward scales, ensuring affinity dominates the policy update. For motif scaffolding, equal weights with different functional forms ensure that global structural recovery dominates early training before seamlessly shifting to precise motif geometry.

## D. Ablation Studies

### D.1. Ablation Studies on Reward Configurations

To evaluate the impact of the composite reward formulation, we conducted ablation experiments by retraining the model across various weight configurations, $(w_{\text{motif}}, w_{\text{global}}, w_{\text{vina}}, w_{\text{div}})$, as summarized in Table 5. The empirical results show that configuration `2801` consistently outperforms both `8201` and `4401#`, validating that our "global-first" optimization strategy facilitated by the flat-top Cauchy function is significantly more effective than merely scaling up local motif weights. Furthermore, the peak performance achieved by our default configuration, `4401`, underscores the necessity of properly balancing these individual reward components. Notably, baseline geometric thresholds ($\delta = 2.0\,\text{Å}, \tau = 1.0\,\text{Å}$) were kept fixed as they represent well-established standards within the protein design community (Yim et al., 2024; Watson et al., 2023).

### D.2. Ablation Studies on IAPO

To isolate the distinct architectural contributions of our proposed framework, we evaluate the individual performance gains attributed to the function-aware reward and the inference-aligned policy optimization (IAPO) mechanism. Our baseline

*Table 5.* Quantitative ablation results across varying reward weight configurations. The 'Configuration' column specifies the weights for $(w_{\text{motif}}, w_{\text{global}}, w_{\text{vina}}, w_{\text{div}})$. For instance, 4401 represents (4.0, 4.0, 0.0, 1.0), which corresponds to our default base setting. The entry 4401# indicates that the global RMSD reward is optimized without applying the flat-top Cauchy function.

| Configuration | Dataset | SR (%) ↑ | mRMSD* (Å) ↓ | RMSD* (Å) ↓ |
|---|---|---|---|---|
| 4401# | FrameF | 34.17 | 1.53 | 3.13 |
| 8201 | FrameF | 29.17 | 1.43 | 3.21 |
| 2801 | FrameF | 35.83 | 1.47 | 3.27 |
| 4401 | FrameF | **37.50** | 1.43 | 3.29 |
| 4401# | RFD | 59.17 | 1.14 | 2.13 |
| 8201 | RFD | 50.00 | 1.20 | 2.14 |
| 2801 | RFD | 59.17 | 1.02 | 1.64 |
| 4401 | RFD | **65.83** | **0.94** | **1.59** |

implementation of standard GRPO utilizes the identical composite reward and sample budget as FIDIA, effectively serving as a direct ablation control. As summarized in our empirical results, the performance gain from the ProteinMPNN baseline to standard GRPO—yielding an increase in success rate from 55.00% to 58.33%—directly quantifies the impact of optimizing for the function-aware composite reward. Furthermore, the subsequent margin of improvement from standard GRPO to FIDIA, which elevates the success rate from 58.33% to 65.83%, rigorously isolates the performance gains achieved explicitly through our inference-aligned policy optimization strategy.

### D.3. Ablation Studies on Temperature

Following established conventions in generative macromolecular design (Wang et al., 2025; Xu et al., 2025), standard reinforcement learning baselines typically employ a training temperature of 1.0 alongside a low inference temperature of 0.1. Because FIDIA explicitly aims to alleviate the training-inference alignment gap, this severe temperature mismatch introduces a distributional shift that inherently compromises optimization stability. To mitigate this while maintaining sufficient exploratory diversity during reinforcement learning, we trained FIDIA at a compromised temperature of 0.3. To ensure a rigorous and unified comparison on the RFD benchmark, we additionally retrained the standard GRPO baseline at this identical 0.3 temperature configuration. As demonstrated in Table 6, while lowering the training temperature yields marginal improvements for standard GRPO, FIDIA consistently and substantially outperforms it across all structural metrics, confirming that our framework's performance gains stem directly from our core policy optimization strategy rather than artifactual temperature tuning.

*Table 6.* Quantitative ablation results across varying training temperature configurations on the RFD benchmark. All configurations utilize an inference temperature of 0.1.

| Model-Temperature | SR (%) ↑ | mRMSD* (Å) ↓ | RMSD* (Å) ↓ |
|---|---|---|---|
| GRPO-1.0 | 58.33 | 1.14 | 1.88 |
| GRPO-0.3 | 59.17 | 1.12 | 1.73 |
| FIDIA-0.3 | **65.83** | **0.94** | **1.59** |

### D.4. Ablation Studies on Inference Sampling Budgets

To evaluate the scalability and computational efficiency of our framework, we investigated the performance of the base ProteinMPNN, standard GRPO, and FIDIA across scaling sequence sample budgets ($N = 8, 16, 64$) during inference. As summarized in Table 7, while increasing the sampling budget systematically elevates the success rate of the ProteinMPNN baseline from 55.00% to 65.83%, relying solely on such test-time scaling becomes computationally prohibitive when evaluation is expensive or time-consuming, such as with Vina docking. In contrast, FIDIA at a minimal budget of $N = 8$ achieves a success rate of 65.83% within 124 seconds, matching the peak performance of ProteinMPNN at $N = 64$ which demands 790 seconds, thereby achieving a over six-fold reduction in computational overhead. Furthermore, FIDIA consistently maintains a robust performance margin over the standard GRPO baseline across all evaluated $N$ budgets, validating that our framework successfully shifts the stochastic, resource-intensive "generate-and-filter" convention into a

highly directed, efficiency-driven generation pipeline.

*Table 7.* Quantitative comparison across varying inference sampling budgets ($N$) on the RFD benchmark.

| Model-$N$ | SR (%) ↑ | mRMSD* (Å) ↓ | RMSD* (Å) ↓ |
|---|---|---|---|
| MPNN-8 | 55.00 | 1.25 | 2.26 |
| GRPO-8 | 58.33 | 1.14 | 1.88 |
| FIDIA-8 | 65.83 | 0.94 | 1.59 |
| MPNN-16 | 60.83 | 1.12 | 1.93 |
| GRPO-16 | 65.00 | 1.03 | 1.66 |
| FIDIA-16 | 70.00 | 0.88 | 1.41 |
| MPNN-64 | 65.83 | 0.96 | 1.50 |
| GRPO-64 | 71.67 | 0.84 | 1.34 |
| FIDIA-64 | **76.67** | **0.73** | **1.16** |

