# OpenReview forum: "FIDIA: Function-Informed Sequence Design via Inference-Aligned Policy Optimization"
_ICML.cc/2026/Conference — ICML 2026 spotlight_

### Official Review · Reviewer_UoAw · 2026-03-01

**Soundness:** 3
**Presentation:** 3
**Significance:** 4
**Originality:** 2
**Overall Recommendation:** 5
**Confidence:** 4

**Summary:**

The paper presents FIDIA, a reinforcement learning framework designed to improve computational protein sequence design. Traditional inverse folding models (e.g., ProteinMPNN) are trained via Maximum Likelihood Estimation to recover native sequences. This fails to account for sequence-structure degeneracy and complex functional requirements. FIDIA explicitly formulates the training objective to maximize the expected maximum reward of a candidate batch. It integrates functional constraints architecturally by appending a binary task-aware mask to residue nodes (distinguishing fixed motifs from redesignable scaffolds) , and functionally through a hand-crafted composite reward function that balances structural metrics (RMSD via ESMFold) and functional metrics (binding affinity via AutoDock Vina). The framework is evaluated against standard base models (ProteinMPNN, ESM-IF, LigandMPNN) and RL/BoN baselines (GRPO, MultiDPO, $BoN_{top1}$) on general motif scaffolding, vaccine epitope scaffolding, and affinity-enhancing enzyme design. FIDIA consistently achieves higher success rates and improved binding scores.

**Compliance With Llm Reviewing Policy:**

Affirmed.

**Final Justification:**

I raised my rating from weak accept to accept with a slightly lower confidence. I have reviewed both the paper and the authors' rebuttal on the issues raised by other reviewers. The rebuttal has largely addressed my issues. I hope the authors could update the manuscript accordingly so that their ideas of functional informed design and IAPO widely benefit the community.

**Key Questions For Authors:**

- Can you provide a quantitative comparison of the wall-clock training time and memory footprint between FIDIA, standard ProteinMPNN, and the GRPO/MultiDPO baselines?
- AutoDock Vina is known to have a loose correlation with experimental binding affinities and can be susceptible to adversarial exploitation by generative models. Did you observe any instances of "reward hacking" (e.g., structurally implausible side-chain packing that artificially deflates the Vina score), and how does FIDIA mitigate this?
- In Eq. 38, you introduce task-specific reward weights (e.g., $w_{vina}=14.0$). How were these specific scalar values determined? Have you conducted ablation studies to evaluate the policy's sensitivity to these hyperparameters?
- In Table 4, FIDIA shows degraded structural metrics (mRMSD and RMSD) compared to the LigandMPNN baseline in pursuit of a better Vina score. At what point does the structural deviation become unacceptable for a viable enzyme, and how do you calibrate the reward to prevent the model from breaking the scaffold to satisfy the Vina oracle?

**Limitations:**

Yes.

**Strengths And Weaknesses:**

Strengths:
- The mathematical formulation of the BoN policy gradient is creative. Deriving an unbiased, variance-reduced estimator (Proposition 3.2 and 3.3) that utilizes the entire sub-level set as a contrastive baseline is a novel and elegant contribution. The derivations are solid, leveraging established statistical principles like order statistics.
- Aligning the training objective with the inference-time screening protocol is a practical advancement for the AI-for-science community. By shifting the probability mass toward the high-fitness tail rather than the mean, the model is much better suited for actual wet-lab discovery pipelines.
- The experimental design contains robust elements. The authors thoughtfully cross-validate their ESMFold-optimized sequences using AlphaFold2. This demonstrates that the performance gains are real structural improvements and not merely the policy overfitting to the ESMFold oracle.


Weaknesses:
- The reward design is arguably the weakest empirical point. The composite reward relies on highly specific, hand-tuned hyperparameter weights (e.g., $w_{motif}=3.0$, $w_{global}=4.0$, $w_{vina}=14.0$, $w_{div}=1.0$). Of course the reward design is a tricky problem even in the general RL field, but there is no theoretical justification or ablation study provided for e.g., why the Vina score is weighted 14 times heavier than diversity, or how sensitive the model is to these heuristic choices.
- The reliance on AutoDock Vina for the enzyme affinity task. Vina scores have limited correlation with true in vitro kinetics. Furthermore, as seen in Table 4, FIDIA trades off structural accuracy (mRMSD 0.64, RMSD 1.29) to achieve a better Vina score (-7.21) compared to the base LigandMPNN (mRMSD 0.56, RMSD 1.13). Optimizing a policy to push Vina scores at the expense of structural degradation might lead to over-optimization against an imperfect physical proxy rather than generating genuinely better enzymes. From my perspective, finding a suitable physical proxy (depending on the task) is very important for the overall framework.
- The framework requires folding and docking evaluations inside the training loop for $K$ sequences per backbone per step. The paper lacks a concrete discussion or ablation on the wall-clock time and memory overhead compared to standard MLE training or simpler RL baselines
- Justification of the "Originality" score: both the function-informed seq design (motif-based) and the RL/Preference Optimization are not new, in the area of the AI4Science, especially the biomolecules design. The authors may include another module of concurrent works using RL to optimize the Protein/biomolecule designs.

References:
- Reinforcement Learning for Sequence Design Leveraging Protein Language Models
- Advancing Protein Design via Multi-Agent Reinforcement Learning with Pareto-Based Collaborative Optimization
- Accelerating protein engineering with fitness landscape modelling and reinforcement learning
- What Does Protein Language Model Learn During Reinforcement Learning?
- Model-Based Reinforcement Learning for Biological Sequence Design
- Preference optimization of protein language models as a multi-objective binder design paradigm
- Improving Protein Sequence Design through Designability Preference Optimization

---

> ### Author Rebuttal · Authors · 2026-03-31
>
> We appreciate your thorough review and helpful comments. We have carefully addressed your concerns below and welcome any additional feedback.
>
> > Q1&W3: wall-clock training overhead
>
> We appreciate you raising this practical consideration (also noted by Reviewer N5cm). Because FIDIA and the RL baselines use the exact same sampling budget, **their wall-clock time and memory footprints are practically identical**. In all cases, the computational cost is overwhelmingly dominated by the shared ESMFold and VinaDock evaluations. While these oracles make RL inherently more time-consuming than standard MLE, we actively mitigated this bottleneck via highly parallelized computations, reducing the total training time to approximately 14 hours on four NVIDIA A100 GPUs.
>
> > Q2: potential reward hacking of the Vina
>
> Thank you for raising this important point. By optimizing in sequence space, FIDIA requires every generated sequence to be folded by ESMFold prior to vina evaluation. ESMFold acts as a strong biological prior, regularizing the outputs into a physically realistic distribution. This naturally prevents reward hacking, as the model is constrained from generating implausible side-chain packing to exploit the Vina score. Furthermore, the high mpLDDT and pLDDT metrics we report (with 85 as a common baseline [1]) demonstrate ESMFold's strong confidence in the predicted folds, implicitly confirming that the generated structures remain physically realistic and structurally sound.
>
> | Method     | Vina  | SR    | mpLDDT* | pLDDT* |
> | ---------- | ----- | ----- | ------- | ------ |
> | LigandMPNN | -6.53 | 90.74 | 92.61   | 91.97  |
> | GRPO       | -7.06 | 75.93 | 89.46   | 88.72  |
> | FIDIA      | -7.21 | 83.33 | 87.73   | 87.68  |
> > Q3&W1: ablation studies
>
> We agree this analysis is highly valuable. Our scalar weights are derived from the distinct optimization dynamics of each task. **For enzyme scaffolding,** optimizing Vina affinity is much harder than improving structural metrics. Thus, we set the Vina weight (14) to twice the combined structural weights (3+4) to balance the reward scales, ensuring affinity dominates the policy update. **For motif scaffolding,** equal weights with different functional forms ensure that global structural recovery dominates early training before seamlessly shifting to precise motif geometry.
>
> Finally, we ran new ablations on RFD benchmark by retraining the model with various weight configurations for the composite reward. Results show that 2801 outperforms both 8201 and 4401#, demonstrating that our 'global-first' optimization strategy via the flat-top Cauchy function is more effective than simply increasing motif weights. Furthermore, the peak performance of 4401 highlights the importance of balancing these reward components. Due to space limit, please refer to our response to Reviewer N5cm (Q1) for detailed weight ablation results.
>
> | weights | SR    | mRMSD* | RMSD* |
> | ------- | ----- | ------ | ----- |
> | 4401#   | 59.17 | 1.14   | 2.13  |
> | 8201    | 50    | 1.2    | 2.14  |
> | 2801    | 59.17 | 1.02   | 1.6   |
> | 4401    | 65.83 | 0.94   | 1.59  |
>
> *Note: Numeric suffixes denote the weight configurations for ($w_{motif}$, $w_{global}$, $w_{vina}$, $w_{div}$). For example, 4401 represents (4.0, 4.0, 0.0, 1.0), which is the default setting used in our original paper. 4401# indicates the global RMSD is optimized without the flat-top Cauchy function.*
>
> > Q4&W2: Trade-off Between Vina Proxy and Structure
>
> We calibrated our structural rewards using Flat-top Cauchy functions with biologically recognized thresholds: 1.0 Å for mRMSD and 2.0 Å for global RMSD. Structures within these bounds are considered highly accurate. Therefore, our optimization of the Vina score occurs with this structural regularization, which also effectively mitigates the risk of reward hacking against the Vina oracle mentioned in Q2. As FIDIA's final metrics (mRMSD 0.64, RMSD 1.29) remain well within these viable biological limits, we achieve enhanced affinity without compromising structural integrity.
>
> > W4: Justification of the "Originality" score
>
> We respectfully clarify our originality, which stems from three advances: (1) addressing the function-agnostic limitations of inverse folding rather than just optimizing physical metrics; (2) exposing the overlooked training-inference gap in the widely adopted BoN protocol for biomolecule design; and (3) introducing a novel clean GRPO-style algorithm to maximize expected maximum rewards. Since existing RL4bio frameworks are either incompatible with tackling the aforementioned challenges or closed-source [2,3,4], we will open-source our codebase to benefit the community.
> 1. Dauparas et al. LigandMPNN. Nat Methods (2025).
> 2. Hou et al. MoMPNN. arXiv:2603.06748 (2025).
> 3. Wang et al. ProteinZero: Self-Improving Protein Generation. arXiv:2506.07459 (2025).
> 4. Xu et al. Protein Inverse Folding From Structure Feedback. arXiv:2506.03028 (2025).

---

> > ### Author Rebuttal · Reviewer_UoAw · 2026-04-02
> >
> > For the enzyme task specifically, have you conducted a reward weight ablation analogous to the one presented for the RFD scaffolding benchmark? The w_vina=14 choice is the most aggressive weighting in the paper and warrants direct sensitivity analysis.
> >
> > You argue ESMFold prevents reward hacking, but ESMFold confidence (pLDDT) is a self-referential metric. Have you evaluated any independent structural quality measure on FIDIA-designed sequences to rule out exploitation of the oracle?
> > Among the concurrent RL-for-protein-design references the reviewer cited, which are most methodologically proximate to FIDIA, and what are the concrete technical distinctions?

---

> > > ### Author Response · Authors · 2026-04-05
> > >
> > > Thank you for your continued time and effort in reading our previous response. Regarding your follow-up questions, please find our detailed responses below:
> > >
> > > > Ablation study on the enzyme task
> > >
> > > Thank you for highlighting this point, and we apologize that Vina docking's computational overhead and time constraints limited our initial ablation to the scaffolding task. We have now added the enzyme task sensitivity analysis.
> > > As shown below, a 3-4-4-1 configuration yields slightly better structural fidelity but results in a degraded Vina metric compared to our chosen 3-4-14-1 setting. We attribute this degradation to affinity losing its dominance in the policy update. Specifically, we observed that during training, structural rewards quickly converge to ~0.8-0.9, while the Vina reward plateaus near 0.4. This discrepancy is rooted in our normalization strategy: while mapping an optimal Vina score (-12) to a maximum reward of 1 is reasonable, it consequently reduces a strong intermediate score (-7.5) to a mere 0.5. Given the inherent difficulty of normalizing disparate metrics to a uniform 0-1 interval while balancing multimodal objectives, setting the Vina weight to 14—twice the combined structural weights (3+4)—was a necessary design choice to balance the reward scales and ensure that binding affinity successfully drives the optimization.
> > >
> > > | Method         | Vina  | mRMSD* | RMSD* |
> > > | -------------- | ----- | ------ | ----- |
> > > | FIDIA-3-4-4-1  | -6.95 | 0.62   | 1.21  |
> > > | FIDIA-3-4-14-1 | -7.30 | 0.64   | 1.29  |
> > >
> > > *Note: Numeric suffixes denote the weight configurations for ($w_{motif}$, $w_{global}$, $w_{vina}$, $w_{div}$). For example, 3-4-14-1 represents (3.0, 4.0, 14.0, 1.0), which is the default setting used in our original paper.*
> > >
> > > > Independent Structural Validation
> > >
> > > We appreciate the reviewer’s insightful point that independent cross-validation of structural quality is indeed essential to rule out oracle exploitation. As briefly explored in Table 2 of our original manuscript, we have already conducted cross-validation on the motif-scaffolding benchmark using AlphaFold2. This provided an independent assessment of structural fidelity, confirming that the improvements achieved by our model reflect actual gains in design quality rather than an artifact of the ESMFold oracle. Furthermore, to fully address the concern regarding potential reward-hacking in the Vina benchmark, we have now used AlphaFold2 to predict the structures of the FIDIA-designed sequences. The evaluation results are presented below, demonstrating no observable signs of reward hacking, even under the more stringent confidence threshold of 90 [1].
> > >
> > > | Method     | Vina  | SR    | mpLDDT* | pLDDT* |
> > > | ---------- | ----- | ----- | ------- | ------ |
> > > | LigandMPNN | -6.53 | 87.04 | 96.61   | 95.86  |
> > > | GRPO       | -7.06 | 81.48 | 95.36   | 94.59  |
> > > | FIDIA      | -7.30 | 85.19 | 94.70   | 94.15  |
> > >
> > > 1. https://alphafold.com/faq
> > >
> > > > Comparison with Concurrent Methods
> > >
> > > We appreciate the reviewer’s insights into RL for biomolecule design and apologize for not detailing this comparison earlier. Among the cited works, Xue et al. and Cao et al. [7,4] are methodologically closest to FIDIA, yet concrete technical distinctions remain. **First**, FIDIA distinctly optimizes the BoN policy, fundamentally shifting the optimization objective compared to these methods. **Second**, regarding task focus, while these methods focus on structural consistency within standalone inverse folding, FIDIA addresses the function-agnostic issue of the inverse folding phase (post-structure generation) when integrated into an end-to-end pipeline, realized through function-informed composite rewards.
> > >
> > > The remaining references address tasks distinct from inverse folding: [6] target de novo sequence generation without prior structural conditioning, and [1,3] focus on mutation optimization. Additionally, general RL techniques like DPO and GRPO [2,4,6,7] are already included as baselines, and approaches requiring training reward model or value networks [1,3,5] fall outside our framework's scope. We will incorporate these works into the Relate work section of the final manuscript.
> > >
> > > 1. Subramanian et al. Reinforcement Learning for Sequence Design Leveraging Protein Language Models. (2024).
> > > 2. Zhu et al. Advancing Protein Design via Multi-Agent Reinforcement Learning with Pareto-Based Collaborative Optimization. (2026).
> > > 3. Sun et al. Accelerating protein engineering with fitness landscape modelling and reinforcement learning. (2025).
> > > 4. Cao et al. From Supervision to Exploration: What Does Protein Language Model Learn During Reinforcement Learning? (2025).
> > > 5. Angermueller et al. Model-based reinforcement learning for biological sequence design. (2019).
> > > 6. Mistani et al. Preference optimization of protein language models as a multi-objective binder design paradigm. (2024).
> > > 7. Xue et al. Improving Protein Sequence Design through Designability Preference Optimization. (2025).

---

### Official Review · Reviewer_N5cm · 2026-03-11

**Soundness:** 3
**Presentation:** 3
**Significance:** 3
**Originality:** 3
**Overall Recommendation:** 5
**Confidence:** 4

**Summary:**

FIDIA addresses two limitations of inverse folding models like ProteinMPNN: (1) they are function-agnostic, optimizing P(sequence|backbone) rather than P(sequence|backbone, function), and (2) their MLE training objective is misaligned with the Best-of-N (BoN) inference protocol used in practice. The key contribution is an analytical policy gradient for the BoN objective that decomposes into a term pushing mass toward the peak performer and a term reinforcing the supporting sub-level distribution. A U-statistic-based estimator achieves provably lower variance than naive partitioning. The method is wrapped in a GRPO-like framework with rank-based advantages and PPO clipping. On motif scaffolding benchmarks, FIDIA raises success rate from 34.17% to 37.50% (FrameF) and from 58.33% to 65.83% (RFD), with additional experiments on vaccine epitope scaffolding and affinity-enhancing enzyme design.

**Compliance With Llm Reviewing Policy:**

Affirmed.

**Final Justification:**

FIDIA derives an analytical Best-of-N policy gradient with provable variance reduction (U-statistic estimator via Rao-Blackwellization) and applies it to close the training-inference gap in inverse folding models. The core theory (Propositions 3.1–3.3) is clean and correct, and the problem — MLE training misaligned with BoN inference — is real and underappreciated in protein design.

  The rebuttal materially strengthened the paper. The N-scaling comparison was the most impactful new result: FIDIA at N=8 matches the base MPNN at N=64 in success rate (65.83%) at ~6× less compute, and FIDIA consistently outperforms GRPO at every N tested (8, 16, 64). This shifts the story from "modest absolute gains" to "equivalent performance at a fraction of the sampling budget" — a much
  stronger practical argument. The reward weight ablation (4401 vs 4401# vs 8201 vs 2801) confirmed that the flat-top Cauchy function matters and that naive motif weight increases underperform the global-first strategy. Training overhead was clarified as identical to standard GRPO (~14 hours on 4×A100), removing a practical concern.

  The authors also made a fair case on W2 (adaptation vs new paradigm): prior BoN work in LLMs focuses on distilling a BoN-induced policy, while FIDIA derives the exact analytical gradient — a stronger mathematical contribution than a direct transfer. I still view the conceptual territory as shared with the LLM alignment literature, but the specific technical contribution goes beyond what
  those prior works provide.

  The hidden text concern is resolved as ICML program committee canary text. The remaining weakness — moderate absolute improvements on FrameFlow (~3.3%) — is mitigated by the 7.5% RFD gain and the efficiency argument from the N-scaling table. A strong paper that earns acceptance.

**Key Questions For Authors:**

1. How sensitive is performance to the reward function design (kernel shapes, weight ratios, thresholds)? An ablation varying these would strengthen the paper.
2. What is the wall-clock training overhead relative to standard GRPO? The K=16 trajectory sampling with ESMFold/Vina evaluation per step seems expensive.
3. Have you compared against simply running BoN at inference time with a larger N, without any training-time alignment? This would isolate the value of training-time BoN optimization vs. just sampling more.

**Limitations:**

Yes. The Discussion section (Sec. 5) lists three key limitations: (i) computational overhead from iterative batch sampling, (ii) reliance on surrogate oracle fidelity, and (iii) need for retraining per task. These are fair self-assessments. The paper could additionally discuss the circularity of using ESMFold as both training oracle and evaluation metric (partially mitigated by the AF2 cross-validation).

**Strengths And Weaknesses:**

## Strengths

- **Clean BoN policy gradient derivation.** Proposition 3.1 (log-CDF gradient as a conditional expectation over the sub-level set) is an elegant result, and the U-statistic estimator (Prop. 3.2) with provable variance reduction via Rao-Blackwellization is the strongest theoretical contribution. Proposition 3.3 showing that mean-centering the rank weights implicitly performs baseline subtraction is a nice practical insight.

- **Well-motivated problem.** The training-inference mismatch (MLE optimizes average quality; practitioners select the best-of-N) is a real and underappreciated issue in protein design pipelines. The paper clearly articulates why optimizing the expected maximum is more aligned with practical workflows than the expected average.

- **Thorough experiments.** Cross-validation with AlphaFold2 (Table 2) addresses the concern of overfitting to the ESMFold oracle. Two backbone generators (FrameFlow, RFdiffusion), two tasks (motif scaffolding, enzyme affinity), and six baselines including both RL (GRPO, MultiDPO) and BoN distillation (BoNtop1, BoNtopk) make for a comprehensive evaluation.

---

## Weaknesses

- **Moderate improvements.** On the primary benchmark (FrameF), the SR gain is 37.50% vs 34.17% — about 3 percentage points. Vaccine scaffolding: 27.09% vs 25.45%. Enzyme Vina: -7.21 vs -7.06. These are consistent improvements but the margins are not large, especially given the added complexity.

- **Adaptation rather than new paradigm.** BoN-aware optimization is well-studied in LLM alignment (Chow et al., 2024; Verdun et al., 2025). The paper's claim that training-inference misalignment is "unexplored within the protein design pipeline" is fair, but the core technique is a transfer from the LLM domain. The protein-specific contribution is primarily the reward design and application.

- **Reward design is hand-crafted with no sensitivity analysis.** The composite reward uses Cauchy kernels, flat-top functions, and task-specific weight tuning (wmotif=4.0, wglobal=4.0, etc.). No ablation is provided on these choices. How sensitive is performance to the kernel shapes, thresholds (delta=2.0A, tau=1.0A), or weight ratios?

---

> ### Author Rebuttal · Authors · 2026-03-31
>
> Thank you for your valuable comments. We have addressed your concerns below and welcome any further feedback.
>
> > Q3: larger N performance
>
> To address this, we evaluated the base MPNN with increasing inference budgets (N=8, 16, 64). While scaling N improves MPNN's success rate (55.00% → 60.83% → 65.83%), relying solely on this approach becomes computationally prohibitive. In contrast, FIDIA at N=8 (124s) achieves comparable performance to MPNN at N=64 (790s). Furthermore, we compared FIDIA against the GRPO baseline across different N budgets, where FIDIA consistently maintains its superiority. This dual advantage in performance and efficiency demonstrates how FIDIA shifts the stochastic "generate-and-filter" design routine to a directed, demand-driven generation pipeline. Detailed results are provided in the following table:
>
> | model-N  | SR    | mRMSD* | RMSD* |
> | -------- | ----- | ------ | ----- |
> | MPNN-8   | 55.00 | 1.25   | 2.26  |
> | GRPO-8   | 58.33 | 1.14   | 1.88  |
> | FIDIA-8  | 65.83 | 0.94   | 1.59  |
> | MPNN-16  | 60.83 | 1.12   | 1.93  |
> | GRPO-16  | 65.00 | 1.03   | 1.66  |
> | FIDIA-16 | 70.00 | 0.88   | 1.41  |
> | MPNN-64  | 65.83 | 0.96   | 1.50  |
> | GRPO-64  | 71.67 | 0.84   | 1.34  |
> | FIDIA-64 | 76.67 | 0.73   | 1.16  |
>
> > Q1&W3: Missing critical ablations
>
> We appreciate this valuable feedback, which was also raised by Reviewers kzuc and UoAw. To address this, we ran new ablations by retraining the model with various weight configurations for the composite reward. Results show that 2801 outperforms both 8201 and 4401#, demonstrating that our 'global-first' optimization strategy via the flat-top Cauchy function is more effective than simply increasing motif weights. Furthermore, the peak performance of 4401 highlights the importance of balancing these reward components. Please note that we omitted ablations on the thresholds (delta=2.0A, tau=1.0A), as these are established community standards[1,2].
>
> | weights-dataset | SR    | mRMSD* | RMSD* |
> | --------------- | ----- | ------ | ----- |
> | 4401#-FrameF    | 34.17 | 1.53   | 3.13  |
> | 8201-FrameF     | 29.17 | 1.43   | 3.21  |
> | 2801-FrameF     | 35.83 | 1.47   | 3.27  |
> | 4401-FrameF     | 37.5  | 1.43   | 3.29  |
> | 4401#-RFD       | 59.17 | 1.14   | 2.13  |
> | 8201-RFD        | 50    | 1.2    | 2.14  |
> | 2801-RFD        | 59.17 | 1.02   | 1.6   |
> | 4401-RFD        | 65.83 | 0.94   | 1.59  |
>
> *Note: Numeric suffixes denote the weight configurations for ($w_{motif}$, $w_{global}$, $w_{vina}$, $w_{div}$). For example, 4401 represents (4.0, 4.0, 0.0, 1.0), which is the default setting used in our original paper. 4401# indicates the global RMSD is optimized without the flat-top Cauchy function.*
> 1. Yim et al. Improved motif-scaffolding with SE(3) flow matching. arXiv.2401.04082 (2024).
> 2. Watson et al. RFdiffusion. Nature (2023).
>
> > Q2: wall-clock training overhead
>
> Thank you for raising this practical consideration. We would like to clarify that FIDIA does not introduce any additional training overhead compared to standard GRPO. By reformulating our optimization into a GRPO-style objective, both methods require the same training time and share the same computational bottleneck: evaluating initial rewards for the sampled trajectories. Although running ESMFold and VinaDock at each step is inherently time-consuming, we mitigate this overhead through parallelization during training, reducing the total wall-clock time to approximately 14 hours on 4 NVIDIA A100 GPUs.
>
> > W1: Moderate improvements.
>
> Because FIDIA focuses specifically on sequence design, our absolute success rate is inherently bounded by the quality of the backbones produced by the structure generative models. Therefore, we evaluate on two distinct backbone generators (FrameF and RFD) equally, without designating either as 'primary'. While the gain on FrameF is indeed ~3.3%, FIDIA achieves a much larger 7.5% improvement (58.33% to 65.83%) on the RFD dataset. We believe these consistent gains across diverse backbone distributions as well as multiple real-world tasks strongly validate our method's robust effectiveness.
>
> > W2: Adaptation rather than new paradigm.
>
> Thank you for highlighting the cross-domain relevance of our work. We would like to respectfully clarify that our core technique was developed independently, rather than being a direct transfer from the LLM domain. Existing reaserches about BoN in LLMs[1,2] primarily focus on finetuning models to mimic a BoN-induced policy, lacking a rigorous mathematical framework to efficiently optimize the expected maximum reward. Our work addresses this limitation by deriving the exact analytical BoN policy gradient (Prop 3.1) and an efficient optimization strategy (Props 3.2 and 3.3). While successfully validated in protein design, we believe our principled approach can be readily extended to the LLM domain.
>
> 1. Gui et al. BoNBoN Alignment. arXiv.2406.00832 (2025).
> 2. Amini et al. Variational Best-of-N Alignment. arXiv.2407.06057 (2025).

---

> > ### Author Rebuttal · Reviewer_N5cm · 2026-03-31
> >
> > I have read the author response and appreciate the new experimental data.
> >
> >   Q3 / Larger N. This is the most informative new result. The table showing FIDIA-8 (65.83% SR) matching MPNN-64 (65.83% SR) at ~6x less compute is a compelling efficiency argument that wasn't clearly made in the original paper. The consistent gap between FIDIA and GRPO at every N further validates the BoN-aware training. This table should be in the main paper — it's arguably stronger evidence for the method than the original benchmarks.
> >
> >   Q1 / W3 (Reward sensitivity). The weight ablation addresses my concern. The result that the flat-top Cauchy function matters (4401 vs 4401#) and that a "global-first" strategy outperforms naive motif weight increases is useful. The omission of threshold ablations is acceptable given these are community-standard values.
> >
> >   Q2 (Training overhead). Clarification that FIDIA adds no overhead beyond standard GRPO is helpful. 14 hours on 4xA100 is reasonable.
> >
> >   W1 (Moderate improvements). Fair point that the 7.5% gain on RFD is more substantial than the 3.3% on FrameFlow, and that both backbone generators should be weighted equally. I accept this — the RFD gain is meaningful.
> >
> >   W2 (Adaptation vs new paradigm). I appreciate the clarification that the work was developed independently. The distinction between prior BoN work (distilling a BoN-induced policy) and this paper (deriving the exact analytical gradient for the BoN objective) is a reasonable one. I still view the conceptual territory — optimizing expected maxima rather than expected averages — as shared with the LLM alignment literature, but I accept that the specific mathematical contribution (Prop 3.1–3.3) goes beyond what those prior works provide.
> >
> >   Overall: This is a strong rebuttal. The N-scaling table and reward ablation are substantive additions that directly address my two main empirical concerns. I raise my recommendation from 4 (Weak Accept) to 5 (Accept). The efficiency result (matching 8x larger sampling budgets) makes a much stronger practical case than the original submission, and the reward ablation closes the missing analysis gap. The theoretical contribution (analytical BoN gradient) holds up as more than a straightforward transfer from the LLM domain.

---

> > > ### Author Response · Authors · 2026-04-01
> > >
> > > Thank you for your encouraging feedback and for raising your recommendation to an Accept. We are thrilled that the new experimental data successfully addressed your core concerns. Your insights were truly invaluable in improving our paper, allowing us to better validate the effectiveness of our method and analyze its sensitivity. Per your suggestion, we will include the ablation studies and the N-scaling table in the main text. Thank you again for your time and support!

---

### Official Review · Reviewer_kzuC · 2026-03-13

**Soundness:** 3
**Presentation:** 2
**Significance:** 2
**Originality:** 2
**Overall Recommendation:** 4
**Confidence:** 3

**Summary:**

This paper proposes FIDIA, an RL-based framework for inverse folding sequence design. It addresses two core issues: (1) standard inverse folding models (e.g., ProteinMPNN) ignore functional constraints during training, and (2) standard RL training objectives are misaligned with the Best-of-N (BoN) inference protocol used in protein engineering. FIDIA encodes functional constraints as composite rewards and derives an analytical policy gradient that directly optimizes the BoN objective.

**Compliance With Llm Reviewing Policy:**

Affirmed.

**Final Justification:**

Part of my concerns are addressed, and I decided to accordingly raise my score.

**Key Questions For Authors:**

1. Can you provide ablations with "proposed composite reward + standard GRPO" and "simple reward + BoN alignment" to disentangle the two contributions?
2. How do you justify the temperature discrepancy (0.3 vs 1.0)? Can you provide results under a unified temperature setting?

**Limitations:**

yes

**Strengths And Weaknesses:**

**Strengths**

1. Well-defined problem. The objective mismatch between standard RL's $E[R]$ and BoN inference's $E[\max_i R_i]$ is formalized as an explicit optimization problem, capturing a real gap in the protein design pipeline.
2. Multi-level experimental design. AF2 cross-validation mitigates oracle circularity risk, and scaling experiments over structure sampling budgets M provide comprehensive evaluation.
3. Domain-informed reward design. The Cauchy kernel for motif RMSD, flat-top Cauchy for global RMSD, and intra-batch Hamming distance to prevent mode collapse are well-motivated choices.

**Weaknesses**

1. Gap between theory and algorithm. The precisely derived BoN gradient is ultimately reduced to a PPO-style surrogate objective (Eq. 17) via uniform credit assignment, which is a rather coarse approximation. The only core difference from GRPO is replacing the raw return with $\lambda_i$ as weights.
2. Missing critical ablations. "Function-aware rewards" and "BoN alignment" are evaluated jointly, making it impossible to attribute performance gains to either component individually.
3. FIDIA uses a training temperature of 0.3 while baselines use 1.0. No comparison under unified temperature is provided, nor is the rationale for this discrepancy justified.
4. The framework is tightly bound to the autoregressive architecture and BoN inference. Recent work on masked-diffusion-based inverse folding with MCTS inference-time search (e.g., ProtInvTree) offers potentially greater modeling flexibility and search efficiency. FIDIA's derivations do not transfer to non-autoregressive architectures, limiting long-term applicability.
5. Insufficient baseline comparisons, notably missing comparisons with inference-time search methods such as ProtInvTree.

---

> ### Author Rebuttal · Authors · 2026-03-31
>
> Thank you for your thorough review and insightful comments. We have carefully addressed your concerns below.
>
> > Q1&W2: Missing critical ablations
>
> We appreciate you highlighting this.
> - **Composite reward + standard GRPO:** Our existing GRPO baseline is exactly this ablation, as it uses the identical composite reward and sample budget as FIDIA. Thus, the MPNN to GRPO gain (55.00% to 58.33%) isolates the impact of the function-aware reward, while the GRPO to FIDIA gain (58.33% to 65.83%) isolates the alignment.
> - **Simple reward + BoN alignment:** We completely agree this is a valuable addition (also noted by Reviewers N5cm, UoAw). We ran new ablations on RFD benchmark by retraining the model with simplified composite reward. The results confirm that our "global-first" optimization strategy—driven by the flat-top Cauchy function and global consistency reward—is crucial for optimal performance. Due to space limit, please refer to our response to Reviewer N5cm (Q1) for detailed ablation results.
>
> | model-weights | SR    | mRMSD* | RMSD* |
> | ------------- | ----- | ------ | ----- |
> | FIDIA-4001    | 56.67 | 1.12   | 1.77  |
> | FIDIA-4401#   | 59.17 | 1.14   | 2.13  |
> | FIDIA-4401    | 65.83 | 0.94   | 1.59  |
>
> *Note: Numeric suffixes denote the weight configurations for ($w_{motif}$, $w_{global}$, $w_{vina}$, $w_{div}$). For example, 4401 represents (4.0, 4.0, 0.0, 1.0), which is the default setting used in our original paper. 4401# indicates the global RMSD is optimized without the flat-top Cauchy function.*
>
> > Q2&W3: temperature discrepancy
>
> Following prior works [2,3], our baselines train at 1.0 and evaluate at 0.1. Since FIDIA specifically aims to close the training-inference gap, this severe mismatch would cause a distributional shift which undermine our motivation. Therefore, we train FIDIA at 0.3—a necessary compromise to align with low-temperature inference while maintaining the exploration diversity. For a unified comparison, we trained the GRPO baseline at 0.3. As RFD benchmark shown below, while it improves marginally, FIDIA still substantially outperforms it. This confirms our gains stem directly from our policy optimization, not the temperature.
>
> | Model-Temp | SR    | mRMSD* | RMSD* |
> | ---------- | ----- | ------ | ----- |
> | GRPO-1.0   | 58.33 | 1.14   | 1.88  |
> | GRPO-0.3   | 59.17 | 1.12   | 1.73  |
> | FIDIA-0.3  | 65.83 | 0.94   | 1.59  |
> > W1: Gap between theory and algorithm.
>
> We respectfully clarify that our surrogate objective represents a theoretically grounded alignment, not a 'gap'. While the algorithm reduces to replacing raw returns with rank-based weights, this is not an engineering heuristic, but the exact analytical realization of our BoN gradient derivation (Props 3.1–3.3). We view this clean, GRPO-style formulation as a core strength. Furthermore, while uniform credit assignment is an approximation, it remains a standard and necessary trade-off for sparse-reward tasks requiring full-sequence evaluation [1, 2].
>
> > W4: Generalization to Non-Autoregressive Models
>
> While we initially selected MPNN for its established robustness and suitability for motif-scaffolding tasks, we respectfully clarify that FIDIA's core derivations **readily transfer to non-autoregressive architectures**. To demonstrate this, we integrated MapDiff [4], a mask-prior-guided diffusion model, by formulating its denoising process as a Markovian trajectory that FIDIA can seamlessly optimize, akin to recent RL-FlowMatching works [1].
> Since MapDiff cannot natively incorporate motif conditions, we optimized global RMSD instead of the composite reward and benchmark on RFD—which notably still improved motif consistency, as detailed below.
>
> | Model         | SR    | RMSD* | mRMSD* |
> | ------------- | ----- | ----- | ------ |
> | MapDiff       | 12.5  | 5.22  | 2.03   |
> | MapDiff-FIDIA | 24.17 | 4.12  | 1.55   |
>
> > W5: Insufficient baselines
>
> Since ProtInvTree's official source code is unavailable for reproduction [3], we addressed your concern by benchmarking against AlignInversePro [6], another state-of-the-art inference-time search method. While effective, the massive computational overhead of these methods makes them highly time-consuming (requiring an additional ~50s per structure for sequence inference) and even intractable for Vina-based objectives. The benchmark on RFD is presented below, in which SMC and SVDD denote two distinct inference-time enhancement techniques:
>
> | Model      | SR    | mRMSD* | RMSD* |
> | ---------- | ----- | ------ | ----- |
> | MPNN       | 26.67 | 1.67   | 4.44  |
> | Align-SMC  | 34.21 | 1.52   | 3.36  |
> | Align-SVDD | 39.17 | 1.16   | 2.74  |
> | FIDIA      | 37.5  | 1.43   | 3.29  |
> 1. Liu et al. Flow-GRPO. arXiv:2505.05470 (2025).
> 2. Wang et al. ProteinZero. arXiv:2506.07459 (2025).
> 3. Xu et al. MPNN-DPO. arXiv:2506.03028 (2025).
> 4. Bai et al. Mapdiff. Nat Mach Intell (2025).
> 5. github.com/A4Bio/ProteinInvBench/issues/16
> 6. Uehara et al. AlignInversePro. arXiv:2501.09685 (2025).

---

> > ### Author Rebuttal · Reviewer_kzuC · 2026-04-04
> >
> > Thank you for the rebuttal and the additional experiments. Part of my concerns are addressed, and I have accordingly updated my score.

---

> > > ### Author Response · Authors · 2026-04-05
> > >
> > > Thank you for acknowledging our rebuttal and the additional experiments, and for raising the score. We highly appreciate your constructive feedback.

---

### Official Review · Reviewer_Muxz · 2026-03-17

**Soundness:** 3
**Presentation:** 3
**Significance:** 3
**Originality:** 4
**Overall Recommendation:** 5
**Confidence:** 3

**Summary:**

- Function aware, BoN filtering aware inverse folding
- RL fine-tuning ProteinMPNN with the reward derived from ESMFold and VinaDock.
Proposes objective that circumvents the non-differentiability and sample inefficiency of the max operator

**Compliance With Llm Reviewing Policy:**

Affirmed.

**Final Justification:**

Rebuttal was strong and it reinforced my prior assessment of accept (5).

**Key Questions For Authors:**

See above

**Limitations:**

yes

**Strengths And Weaknesses:**

Strengths
- Clear motivation. Tackles important yet unaddressed problem in evaluation pipeline of protein design.
- Derivation of BoN policy gradient seems theoretically sound and original.
- Strong performance, even compared to other RL/BoN baselines.

Weaknesses
 - Usually motif scaffolding benchmarks require 100 samples per problem and then clustering using FoldSeek. 5 samples used in this paper is limited and those 5 samples can be structurally redundant. To truly assess the model’s generalizability across diverse backbone distributions, 100 samples then clustered set would be better.
- Minor formatting issues: Consistently missing whitespace before citation. Also, use citet not citep (e.g. L55-58 second columns) when directly referencing works e.g. "\citet{xu2025} utilizes structural feedback"...

---

> ### Author Rebuttal · Authors · 2026-03-31
>
> Thank you for your constructive feedback and for recognizing the value of our contributions. Guided by your suggestions, we have carefully addressed the raised concerns.
>
> >W1: Sampling scale and redundancy
>
> Thank you for this constructive suggestion. As briefly explored in Figure 1 of our original manuscript, we have evaluated the pipeline across varying structural sampling budgets up to M=100. This initial analysis demonstrated that FIDIA converges to optimal performance (21 solved targets) at M=50, suggesting that its robust sequence sampling capabilities effectively mitigate the necessity for exhaustive structural sampling. To explicitly address the issue of structural redundancy, we conducted the requested evaluation by sampling 100 backbones per motif and clustering them via FoldSeek (TM-score > 0.5, coverage > 0.8). As the results below confirm, FIDIA maintains its superior performance under this setting.
>
> | Method-dataset   | SR    | mRMSD* | RMSD* | TM-score* | mpLDDT* | pLDDT* |
> | ---------------- | ----- | ------ | ----- | --------- | ------- | ------ |
> | MPNN-FrameF-FS   | 20.79 | 1.64   | 4.48  | 0.69      | 67.54   | 70.77  |
> | GRPO-FrameF-FS   | 29.39 | 1.44   | 3.36  | 0.75      | 68.63   | 72.24  |
> | FIDIA-FrameF-FS  | 31.42 | 1.41   | 3.32  | 0.75      | 69.29   | 72.45  |
> | MPNN-vaccine-FS  | 9.19  | 3.69   | 4.86  | 0.74      | 55.96   | 67.58  |
> | GRPO-vaccine-FS  | 11.55 | 3.33   | 4.11  | 0.78      | 57.5    | 68.45  |
> | FIDIA-vaccine-FS | 12.07 | 3.19   | 3.86  | 0.79      | 58.79   | 70.01  |
>
> >W2: formatting issues
>
> We appreciate you pointing out these formatting issues. All missing whitespaces before citations have been added, and the incorrect uses of `\citep` have been replaced with `\citet` throughout the revised manuscript.

---

> > ### Author Rebuttal · Reviewer_Muxz · 2026-04-02
> >
> > Thank you for extending the evaluation! Looks like a strong paper to me

---

> > > ### Author Response · Authors · 2026-04-03
> > >
> > > Thank you very much for your feedback. We sincerely appreciate your recognition of our work!

---

### Decision · Program_Chairs · 2026-04-30

**Decision:**

Accept (spotlight)

**Comment:**

FIDIA addresses a real  problem in computational protein design: the misalignment between MLE training objectives and the Best-of-N inference protocol widely used in practice. The core theoretical contribution an analytical Best-of-N  policy gradient decomposed into a peak-pushing term and a sub-level set reinforcement term, with a U-statistic estimator achieving provable variance reduction via Rao-Blackwellization - this is technically sound and elegant. Four reviewers assessed the paper, with three recommending accept and one weak accept after the rebuttal, reflecting a strong positive consensus. The rebuttal strengthened the submission, particularly the N-scaling experiment  Remaining concerns  are acknowledged but do not undermine the core contribution. The paper makes a principled and practically valuable contribution to the protein design community .